# Supercooled Liquid Water Cloud observed, analysed and modelled at the Top of the Planetary Boundary Layer above Dome C, Antarctica

**Philippe Ricaud[1], Massimo Del Guasta[2], Eric Bazile[1], Niramson Azouz[1], Angelo Lupi[3], Pierre Durand[4], Jean-Luc Attié[4], Dana Veron[5], Vincent Guidard[1] and Paolo Grigioni[6]**

[1]CNRM, Université de Toulouse, Météo-France, CNRS, Toulouse, France

[2]INO-CNR, Sesto Fiorentino, Italy

[3]ISAC-CNR, Italy

[4]Laboratoire d'Aérologie, Université de Toulouse, CNRS, UPS, Toulouse, France

[5]University of Delaware, Newark, USA

[6]ENEA, Roma, Italy

**Version V03.R2, 6 March 2020**

Submitted to ACPD

## Abstract

A comprehensive analysis of the water budget over the Dome C (Concordia, Antarctica) station has been performed during the austral summer 2018-2019 as part of the Year of Polar Prediction (YOPP) international campaign. Thin (~100-m deep) supercooled liquid water (SLW) clouds have been detected and analysed using remotely sensed observations at the station (tropospheric depolarization LIDAR, microwave radiometer HAMSTRAD, net surface radiation from Baseline Surface Radiation Network BSRN), radiosondes and using satellite observations (CALIOP/CALIPSO) combined with a specific configuration of the Numerical Weather Prediction model: ARPEGE-SH (Action de Recherche Petite Echelle Grande Echelle – Southern Hemisphere). The analysis shows that SLW clouds were present from November to March, with the greatest frequency occurring in December and January when ~50% of the days in summer time exhibited SLW clouds for at least one hour. Two case studies are used to illustrate this phenomenon. On 24 December 2018, the atmospheric planetary boundary layer (PBL) evolved following a typical diurnal variation, which is to say with a warm and dry mixing layer at local noon thicker than the cold and dry stable layer at local midnight. Our study showed that the SLW clouds were observed at Dome C within the entrainment and the capping inversion zones at the top of the PBL. ARPEGE-SH was not able to correctly estimate the ratio between liquid and solid water inside the clouds with the Liquid Water Path (LWP) strongly underestimated by a factor 1000 compared to observations. The lack of simulated SLW in the model impacted the net surface radiation that was 20-30 W m$^{-2}$ higher in the BSRN observations than in the ARPEGE-SH calculations, mainly attributable to the BSRN longwave downward surface radiation being 50 W m$^{-2}$ greater than that of ARPEGE-SH. The second case study takes place on 20 December 2018, when a warm and wet episode impacted the PBL with no clear diurnal cycle of the PBL top. SLW cloud appearance within the entrainment and capping inversion zones coincided with the warm and wet event. The amount of liquid water measured

by HAMSTRAD was ~20 times greater in this perturbed PBL than in the typical PBL. Since
ARPEGE-SH was not able to accurately reproduce these SLW clouds, the discrepancy between
the observed and calculated net surface radiation was even greater than in the typical PBL case,
reaching +50 W m$^{-2}$, mainly attributable to the downwelling longwave surface radiation from
BSRN being 100 W m$^{-2}$ greater than that of ARPEGE-SH. The model was then run with a new
partition function favouring liquid water for temperatures below -20°C down to -40°C. In this
test mode, ARPEGE-SH has been able to generate SLW clouds with modelled LWP and net
surface radiation consistent with observations during the typical case whereas, during the
perturbed case, the modelled LWP was 10 times less than the observations and the modelled
net surface radiation remained lower than the observations by ~50 W m$^{-2}$. Accurately modelling
the presence of SLW clouds appears crucial to correctly simulate the surface energy budget
over the Antarctic Plateau.

## 1. Introduction

Antarctic clouds play an important role in the climate system by influencing the Earth's radiation balance, both directly at high southern latitudes and, indirectly, at the global level through complex teleconnections (Lubin et al., 1998). In Antarctica, there are very few observational stations and most of them are located on the coast, a fact that limits the type and characteristics of clouds observed. Nevertheless, prior studies suggest that cloud properties vary geographically, with a fractional cloud cover around the South Pole of about 50 to 60% in all seasons, and a cloud cover of about 80 to 90% near the coast (Bromwich et al., 2012; Listowski et al., 2019). Based on spaceborne observations, Adhikari et al. (2012) observed that low-level cloud occurrence over the Antarctic Plateau is consistently between 20-50% with the highest values occurring in winter and the lowest values consistently occurring over the Eastern Antarctic Plateau. Furthermore, cloud parameters such as the hydrometeors size and the microphysical structure are also very difficult to retrieve in Antarctica. Nevertheless, some in situ aircraft measurements exist particularly over the Western Antarctic Peninsula (Grosvenor et al., 2012; Lachlan-Cope et al., 2016) and nearby coastal areas (O'Shea et al., 2017) that provide ice mass fraction, concentration and particle size relative to cloud temperature, cloud type and formation mechanism which have provided new insights to polar cloud modelling. These studies also highlighted sea-ice production of Cloud-Condensation Nuclei and Ice Nucleating Particles, which is important in winter both coastally and at Dome C (see e.g. Legrand et al., 2016). Additionally, Grazioli et al. (2017) observed precipitating crystal characteristics at Dumont d'Urville using a combination of ground-based radars, in situ cameras and precipitation sensors, and looked at the role that the katabatic winds play in the formation, modification and sublimation of ice crystals. Over the Antarctic Plateau, where the atmosphere is colder and drier than along the coast, ice crystal clouds are mainly observed with crystal sizes ranging from 5 to 30 μm (effective radius) in the core of the cloud; mixed-phase clouds are

preferably observed near the coast (Listowski et al., 2019) with larger ice crystals and water
droplets (Lachlan-Cope, 2010; Lachlan-Cope et al., 2016; Grosvenor et al., 2012; O'Shea et al.,
2017; Grazioli et al., 2017).

The time and geographical distribution of tropospheric clouds over the Antarctic region

has been recently studied using the raDAR/liDAR-MASK (DARDAR) spaceborne products
(Listowski et al., 2019). The authors determined that clouds are mainly constituted of ice above
the continent. The presence of Supercooled Liquid Water (SLW, the water staying in liquid
phase below 0°C) clouds shows variations according to temperature and sea ice fraction,
decreasing sharply poleward, with an abundance two to three times less over the Eastern
Antarctic Plateau than over the Western Antarctic. The inability of mesoscale high-resolution
models and operational numerical weather prediction models to accurately calculate the net
surface radiation due to the presence of clouds (particularly of SLW clouds) in Antarctica
causes biases of several tens of watt per square meters (Listowski and Lachlan-Cope, 2017,
King et al., 2006, 2015; Bromwich et al., 2013) impacting the radiative budget of the Antarctic
and beyond (Lawson and Gettelman, 2014; Young et al. 2019). The year-long study of mixed-
phase clouds at South Pole with a micropulse LIDAR presented in Lawson and Gettelman
(2014) showed that SLW clouds occur more frequently than observed in earlier aircraft studies,
and are underestimated in models leading to biases in the surface radiation budget. In the present
study, we explore these biases further, moving the focus to the modelling and simultaneous
observations of low-level SLW clouds and surface radiation over the Eastern Antarctic Plateau,
specifically at Dome C.

With the support of the World Meteorological Organization (WMO) World Weather

Research Programme (WWRP), the Polar Prediction Project (PPP) international programme
has been dedicated to the development of improved weather and environmental prediction
services    for    the    polar    regions,    on    time    scales    from    hours    to    seasons
(https://www.polarprediction.net). Within this project, the Year of Polar Prediction (YOPP),
from 2018 to 2019, aims at enabling a significant improvement in environmental prediction
capabilities for the polar regions and beyond, by coordinating a period of intensive observing,
modelling, verification, user-engagement and educational activities. The Water Budget over
Dome C ($H_2O$-DC) project (https://apps3.awi.de/YPP/pdf/stream/52) has been endorsed by
YOPP for studying the water budget by means of ground-based measurements of water (vapour,
solid and liquid) and clouds, by active (backscatter LIDAR) and passive (microwave
radiometer) remote sensing, and operational meteorological analyses. The Dome C (Concordia)
station is located in the Eastern Antarctic Plateau (75°06'S, 123°21'E, 3233 m above mean sea
level, amsl).
$H_2O$-DC concentrates on the Year of Polar Prediction Special Observing Period of
measurements in the Antarctic (SOP-SH), from 16 November 2018 to 15 February 2019.
During this time frame, several instruments have been employed.
1) The $H_2O$ Antarctica Microwave Stratospheric and Tropospheric Radiometer
(HAMSTRAD, Ricaud et al., 2010a) to obtain vertical profiles of temperature and water
vapour, Integrated Water Content (IWC) or precipitable water, and Liquid Water Path (LWP),
with an adjustable time resolution fixed at 60 seconds during the YOPP campaign.
2) The tropospheric depolarization LIDAR (Tomasi et al., 2015) to obtain vertical profiles
of backscattering and depolarization ratio.
These two $H_2O$-DC data sets have been complemented in the present analysis by the 3
following observational datasets.
3) The Baseline Surface Radiation Network (BSRN) net surface radiances at the station.
4) The temperature profiles from radiosondes launched twice daily at the station during
YOPP.
5) The spaceborne observations (backscatter and polarization) from the
CALIOP/CALIPSO LIDAR in the vicinity of the station.
In addition, a specific Antarctic configuration of the global ARPEGE model from Météo-
France (Pailleux et al., 2015) is used to characterize the water budget above Dome C
considering the gas, liquid and solid phases to study the genesis of clouds (ice/liquid).
The aim of the present study is to combine all these observations and simulations in order
to 1) detect the presence of SLW clouds above Dome C, 2) analyse the formation and evolution
of such SLW clouds and 3) estimate the radiative impact of such clouds on the net surface
radiation. We concentrate the analyses on two case studies observed during the YOPP
campaign: one case when the Planetary Boundary Layer (PBL) exhibited a "typical" diurnal
cycle (24 December 2018) and a second case when the diurnal cycle of the PBL was perturbed
by a warm and wet episode (20 December 2018).
The data sets used in our study are presented in section 2. The methodology employed is
explained in section 3. The analyses of the SLW clouds during the typical and the perturbed
PBL periods are detailed in sections 4 and 5, respectively. The observed and modelled impact
of SLW clouds on the surface net radiation is described in section 6. Section 7 includes a
discussion of the results and the conclusion synthesizes the study in section 8.

**2. Datasets**
**2.1. The HAMSTRAD Radiometer**
HAMSTRAD is a microwave radiometer that profiles water vapour ($H_2O$), liquid water and
tropospheric temperature above Dome C. Measuring at both 60 GHz (oxygen molecule line
($O_2$) to deduce the temperature) and 183 GHz ($H_2O$ line), this unique, state-of-the-art
radiometer was installed on site for the first time in January 2009 (Ricaud et al., 2010a and b).
The measurements of the HAMSTRAD radiometer allow the retrieval of the vertical profiles
of $H_2O$ and temperature from the ground to 10-km altitude with vertical resolutions of 30 to 50
m in the PBL, 100 m in the free troposphere and 500 m in the upper troposphere-lower
stratosphere. The time resolution is adjustable and fixed at 60 seconds during the YOPP
campaign. Note that an automated internal calibration is performed every 12 atmospheric
observations and lasts about 4 minutes. Consequently, the atmospheric time sampling is 60
seconds for a sequence of 12 atmospheric measurements and a new atmospheric sequence is
performed after 4 minutes. The temporal resolution on the instrument allows for detection and
analysis of atmospheric processes such as the diurnal evolution of the PBL (Ricaud et al., 2012)
and the presence of clouds and diamond dust (Ricaud et al., 2017). In addition, two other
parameters can be estimated.
1) The Integrated Water Vapour (IWV) or precipitable water (kg m$^{-2}$) obtained by
integrating the absolute humidity profile from the surface to 10 km altitude.
2) The Liquid Water Path (g m$^{-2}$) that gives the amount of liquid water integrated along the
vertical.
IWV has been validated against radiosondes at Dome C between 2010 and 2014 showing a
5-10% wet bias of HAMSTRAD compared to the sondes (Ricaud et al., 2015) that were
uncorrected for sensor heating or time lag effect that may produce a 4% dry bias (Miloshevish
et al., 2006). The 1-σ RMS error on the 7-min integration time IWV is 0.05 kg m$^{-2}$ or ~5%
(Ricaud et al., 2013).
The HAMSTRAD-observed LWP has only been presented when the instrument was
installed at the Pic du Midi station (2877 amsl, France) during the calibration/validation period
in 2008 prior to its set up in Antarctica in 2009 (Ricaud et al., 2010a). Because the instrument
has been designed and developed for measuring water vapour in very dry and cold environments
such as those encountered at the Dome C station all year long, the radiometer functionality is
better adapted for the Dome C site than for the Pic du Midi site. It has not been possible to
validate LWP observations at the Pic du Midi station. The H$_2$O-DC project has thus provided a
unique opportunity to perform such a qualitative validation against LIDAR observations of
SLW.

**2.2. The tropospheric depolarization LIDAR**
A tropospheric depolarization LIDAR (532 nm) has been operating at Dome C since 2008
(see http://lidarmax.altervista.org/englidar/_Antarctic%20LIDAR.php). The LIDAR provides
5-min tropospheric profiles of aerosols and clouds continuously, from 20 to 7000 m above
ground level (agl), with a resolution of 7.5 m.  LIDAR depolarization (Mishchenko et al., 2000)
is a robust indicator of non-spherical shape for randomly oriented cloud particles. A
depolarization ratio below 10% is characteristic of SLW clouds, while higher values are
produced by ice particles. The possible ambiguity between SLW clouds and oriented ice plates
is avoided at Dome C by operating the LIDAR 4° off-zenith (Hogan and Illingworth, 2003).
The LIDAR observations at Dome C have already been used to study the radiative properties
of water vapour and clouds in the far infrared (Palchetti et al., 2015). As a support to LIDAR
data interpretation, time-lapse webcam videos of local sky conditions are also collected.

**2.3. The BSRN Network**
The BSRN sensors at Dome C are mounted at the Astroconcordia/Albedo-Rack sites, with
upward and downward looking, heated and ventilated standard Kipp&Zonen CM22
pyranometers and CG4 pyrgeometers providing measurements of hemispheric downward and
upward broadband shortwave (SW, 0.3–3 μm) and longwave (LW, 4–50 μm) fluxes at the
surface, respectively. These data are used to retrieve values of net surface radiation (defined as
the difference between the downward and upward fluxes). All these measurements follow the
rules of acquisition, quality check and quality control of the BSRN (Driemel et al., 2018).

**2.4. Radiosondes**

Vertical temperature and humidity profiles have been measured on a daily basis at Dome C since 2005, employing RS92 Vaisala radiosondes. The radiosonde data were taken using the standard Vaisala evaluation routines without any correction of sensor heating or time lag effect. The sondes are known to have a cold bias of 1.2 K from the ground to about 4 km altitude (Tomasi et al., 2011 and 2012) and a dry bias of 4% on IWV (Miloshevish et al., 2006), mainly between 630 and 470 hPa, with a correction factor for humidity varying within 1.10–1.15 for daytime (Miloshevish et al., 2009). During YOPP and the two case studies, launches were performed twice per day at 00:00 and 12:00 UTC.

**2.5. CALIOP on board CALIPSO**

Orbiting at 705-km altitude, the CALIPSO (Cloud Aerosol Lidar and Infrared Pathfinder Satellite Observations) mini-satellite has been observing clouds and aerosols since 2006 to better understand the role of clouds and aerosols in climate. To accomplish this mission, the CALIPSO satellite is equipped with a LIDAR, a camera and an infrared imager (Winker et al., 2009). CALIOP (Cloud-Aerosol LIdar with Orthogonal Polarization) is a dual-wavelength (532 and 1064 nm) backscatter LIDAR. It provides high-resolution vertical profiles of clouds and aerosols along the orbit track (Young et al., 2009). We have used version V3.40 data retrieved from https://www-calipso.larc.nasa.gov/.

**2.6. The ARPEGE-SH Model**

A special Antarctic configuration of the operational global model ARPEGE was used for the YOPP SOP-SH period (16/11/2018–15/02/2019). This configuration named ARPEGE-SH is based on the operational global model used for Numerical Weather Prediction (NWP)

ARPEGE (Pailleux et al., 2015), but with its highest horizontal resolution centred over Dome
C instead of over France, as set up in ARPEGE. A 4D variational (4DVar) assimilation was
performed every 6 h. The meteorological analyses were given by the ARPEGE-SH system
together with the 24-hour forecasts at the node the closest to the location of Dome C. Two
analyses at 00:00 and 12:00 UTC were represented in the present study together with hourly
forecasts initialized by the two analyses from 01:00 to 11:00 and from 13:00 to 24:00 UTC,
respectively. The horizontal resolution during the SOP-SH period was 7.5 km at Dome C.  The
vertical resolution during the SOP-SH period was constituted by 105 vertical levels, the first
one being set at 10 m, with 12 levels below 1 km and 35 levels below 3 km. Several ARPEGE-
SH output parameters were selected for analysis: cloud fraction, ice, water vapour and liquid-
water mixing ratio, temperature, Total Column Ice (TCI, ice integrated along the vertical),
LWP, IWV, and net surface radiation. For each of the model vertical level, the value of the
cloud fraction ranges between 0 and 1 and is defined as the fraction of the cloud within the
model horizontal grid box. The total cloud fraction at each level is a combination between the
resolved cloud, the cloud from the shallow convection and the cloud from the deep convection.
The resolved cloud is based on a pdf function with critical relative humidity profile. The shallow
convection cloud (below 4000 m) is based on the cloud water/ice tendencies computed by the
shallow mass flux scheme with a maximum value at 0.3. For the deep convection, the cloudiness
is computed with the vertical divergence of the precipitation flux. The diurnal variation of the
top of the PBL is calculated by ARPEGE-SH as the level where the turbulence kinetic energy
becomes lower than 0.01 $m^2 s^{-2}$.

**2.7. The NCEP temperature fields**

In order to assess the synoptic state of the atmosphere during the two case studies above

Dome C against the climatological state of the atmosphere in summer over Antarctica, we have
used the temperature fields at 600 hPa from the National Centers for Environmental Prediction
(NCEP) from 2009 to 2019 (Kanamitsu et al., 2002). These are NCEP-Department of Energy
(NCEP/DOE) Atmospheric Model Intercomparison Project (AMIP-II) Reanalysis (Reanalysis-
2) 6-hourly air temperature at 2.5°x2.5° horizontal resolution over the globe.

**2.8. The HYSPLIT back-trajectories**
In order to assess the origin of airmasses associated to the two case studies, ten-day back-
trajectories originated from the Dome C station at 500 and 1000 m above ground level have
been calculated on 20 and 24 December 2018 at 12:00 UTC from the Hybrid Single-Particle
Lagrangian Integrated Trajectory model (HYSPLIT) model (Stein et al., 2015; Rolph et al.,
2017) (https://www.ready.noaa.gov/HYSPLIT.php).

**3. Methodology**
In this article, we present two case studies from the SOP-SH that illustrate the occurrence
of low-level SLW clouds at Dome C. Both cases occurred in December 2018, within 5 days of
each other, which allows direct comparison between the cases without concerns for seasonal
variations in radiation.
The first case study presented was on 24 December 2018 and was representative of a
climatological summer atmosphere in contrast to the second case study (20 December 2018)
when the atmosphere was very different from a climatological summer atmosphere. We have
considered in Figure 1 the temperature fields from the NCEP at 600 hPa to highlight the state
of the atmosphere above Antarctica with a focus over the Dome C station at different periods:
a) decadal average over December-January from 2009 to 2019, b) YOPP average over
December 2018-January 2019, c) daily average over 24 December 2018, d) 20 December 2018
at 00:00 UTC, e) 20 December 2018 at 12:00 UTC, and f) 21 December 2018 at 00:00 UTC.
The climatological summer temperature field at 600 hPa has been calculated by averaging the
December and January data from 2009 to 2019 and the mean synoptic state of the YOPP
campaign during the summer 2018-2019 has been calculated by averaging data from early
December 2018 to end of January 2019. The synoptic state of the first case study was selected
on 24 December 2018 averaged from 00:00 to 24:00 UTC and for the second case study on 20
December 2018 at 00:00 UTC and 12:00 UTC, and on 21 December 2018 at 00:00 UTC. Firstly,
the summer atmosphere during YOPP was very consistent with the decadal climatological state
of the atmosphere both over Antarctica and the Dome C station (temperature less than 245 K).
Secondly, the synoptic state of the atmosphere on 24 December 2018 (1st case study), although
warmer (> 258 K) over some parts of the Antarctic Plateau (60°E-90°E) is, over Dome C,
consistent with the YOPP summer synoptic state and the climatological summer temperatures
of ~246 K. Thirdly, on 20 December 2018 (2nd case study), on tongue of warm air (254-260 K)
originated from the oceanic coast in the sector 0-30°W (00:00 UTC) reaches Dome C 24 hours
later with temperatures increasing from 252 to 256 K, about 10 K greater than on 24 December
2018. Ten-day back trajectories calculated from HYSPLIT (see Figure Supp1) initiated at
Dome at 500 and 1000 m above ground level remain over the Antarctic Plateau on 24 December
2018 (1st case study) whereas are originated to the oceanic coast in the sector 0-30°W on 20
December 2018 (2nd case study). This is consistent with previous studies (Ricaud et al., 2017)
showing that inland-originated air masses bring cold and dry air to Dome C whilst ocean-
originated air masses bring warm and wet air to Dome C.

In the following, we will label the 1st case study on 24 December 2018 as typical case and

the 2nd case study as perturbed case. We will show that, in the typical case, the SLW cloud
occurred over a 24-hour period that was characterized by a typical summertime, diurnal PBL
cycle, where the mixed-layer develops over the course of the day, reaches a quite stable height
and then collapses to the surface toward the end of the day, around 12 UTC (Ricaud et al.,
2012). The first case provides insight into the impact of SLW clouds on the local radiative
fluxes. The perturbed case provides a contrasting situation where the diurnal cycle of the PBL
was perturbed by the sudden arrival of very moist and warm air of oceanic origin (see Ricaud
et al., 2017). We analyse how this episode affected the presence and evolution of SLW clouds
and their influence on the surface energy budget. Note that, in the remaining of the article, the
data will be presented according to their height above ground level (agl) unless explicitly shown
as above mean sea level (amsl).

## 4. Typical diurnal cycle of the PBL

The first case study occurred on 24 December 2018 during a typical diurnal PBL cycle.
All the results are presented in Universal Time Coordinated (UTC) with local time (LT) being
eight hours ahead of UTC (LT = UTC + 8 hr). As described in Ricaud et al. (2012), the typical
summer boundary layer at Dome C is very similar to that described by Stull (1988). Although
sunlight is present throughout the day, the variation in magnitude is enough to allow a stable
boundary layer from 18:00 to 06:00 LT, similar to a stable nocturnal boundary layer. There is
then a transition from a stable boundary layer to a mixed layer around 06:00 LT with the
increase in the solar irradiation, which reaches a maximum around solar noon. Then around
18:00 LT, the stable boundary layer starts to form again, with a quasi-mixed layer about it. The
height of the summertime boundary layer at Dome C typically ranges between 100 and 400 m.
The presence of SLW clouds at the top of the PBL together with the diurnal evolution of the
PBL will be discussed in more detail in the section 7.2.

### 4.1. Clouds

The presence of clouds is highlighted by the LIDAR backscatter and depolarization profiles
shown in Figures 2a and b, respectively. High values of LIDAR backscatter ($\beta > 100\ \beta_{mol}$, with
$\beta_{mol}$ the molecular backscatter) indicate that clouds and/or precipitation are present
intermittently thought the day with some significant differences. First, vertical "stripes" of high
backscatter values are visible from 10 to 400 m height before 10:00 UTC and after 19:00 UTC,
associated with high values of depolarization ratio (> 20 %), characteristic of precipitating ice
crystals. Second, high values of β associated with very low depolarization ratio (< 5 %) occur
within a thin layer of approximately 100-m depth around 500 m from 08:00 to 22:00 UTC, with
some breaks around 11:00 and 19:00-21:00 UTC. From the LIDAR observations, this
combination of high backscatter and low depolarization ratio signifies the presence of a SLW
cloud (Figure 2c).
The NWP model ARPEGE-SH calculates cloud fraction, ice water and liquid water mixing
ratios (kg kg$^{-1}$) for 24 December 2018 (Figures 3a, b and c, respectively). We note that the
outputs from ARPEGE-SH at 00:00 and 12:00 UTC are the analyses and, for the remaining
time, the outputs are the hourly forecasts. ARPEGE-SH predicts the presence of clouds (cloud
fraction > 0.95) for most of the day except around 11:00 and 23:00 UTC (Fig. 3a). Before 12:00
UTC, the cloud is mainly confined between 300 and 600-800 m whilst, after 12:00 UTC, it
spreads from the surface to 800 m. There are also high-level clouds at 2000-3000 m height but
with a cloud fraction between 0.50 and 0.70. The majority of the clouds produced by ARPEGE-
SH are mainly composed of ice crystals (Fig. 3b) with some traces of droplets (Fig. 3c) due to
the model's partitioning between ice and liquid where all condensated water is ice below -20°C.
The liquid water clouds derived from the LIDAR observations are superposed over the SLW
clouds calculated by ARPEGE-SH. The modelled values of liquid water ($\sim$4 10$^{-6}$ g m$^{-3}$) are very
low, far lower than the values of 0.1 g m$^{-3}$ observed for coastal polar stratus clouds (see e.g.
O'Shea et al., 2017; Lachlan-Cope et al., 2016; Young et al., 2016). It is evident that ARPEGE-
SH fails in estimating: 1) the vertical distribution of liquid water (a thin layer is observed around
500 m whereas the modelled cloud layer extends from the surface to 800 m); 2) its temporal

evolution (presence of SLW cloud almost all day long in ARPEGE-SH compared to SLW clouds from 08:00 to 22:00 UTC in the observations); and 3) the liquid vs. ice mixing ratio, the former being in the model several orders of magnitudes lower than the latter, in contrast to the observations.

The diurnal variation along the vertical of the Total Snow Flux (mm day$^{-1}$) calculated by ARPEGE-SH on 24 December 2018 and on 20 December 2018 is shown on Figures Supp2 and Supp3, respectively. On 24 December 2018 (Fig. Supp2), ARPEGE-SH forecasts some solid precipitation between 00:00 and 10:00 UTC from ~500 m agl to the surface consistently with the LIDAR observations (Figs. 2a and b). On 20 December 2018 (Fig. Supp3), ARPEGE-SH calculates trace amounts of solid precipitation close to the surface around 16:00 UTC consistently with the LIDAR observations (Figs. 9a and b). ARPEGE-SH was thus able to forecast solid precipitation during the 2 case studies.

The presence of clouds above the station can also be inferred from vertically-integrated variables such as: 1) TCI calculated by ARPEGE-SH, 2) LWP from HAMSTRAD and ARPEGE-SH, and 3) IWV from HAMSTRAD and ARPEGE-SH (Figures 4a, b and c, respectively). The ARPEGE-SH TCI on 24 December 2018 (Fig. 4a) oscillates between 10 and 30 g m$^{-2}$ except around 12:00 UTC when a clear minimum occurs (~3 g m$^{-2}$), underscoring the fact that ARPEGE-SH obtains ice clouds for the entire day, except at 12:00 UTC. The HAMSTRAD LWP shows an obvious increase from ~1.0 to ~2.0-3.0 g m$^{-2}$ when the presence of SLW cloud is indicated by LIDAR observations (Fig. 4b). The ARPEGE-SH LWP is, on average, $10^3$ times lower than that observed by HAMSTRAD, highlighting the fact that ARPEGE-SH misrepresents features of the SLW clouds over Dome C. The 1-σ RMS error on the 1-min integration time for the HAMSTRAD LWP can be estimated to be ~15%. Based on the comparisons between the HAMSTRAD LWP and the LIDAR observations of SLW clouds during the YOPP campaign, we can estimate that the LWP bias is about 1.0 g m$^{-2}$. We cannot

rule out that these biases might also be related in part to differences in the observation
wavelengths employed (submicrons for the LIDAR and microwaves for HAMSTRAD) that
could favour large particles (HAMSTRAD) against small particles (LIDAR). Biases might also
be due to the observing geometry that differs between the LIDAR (close to zenith viewing) and
HAMSTRAD (atmospheric scans at 10 angles from zenith to ~3° elevation). HAMSTRAD and
ARPEGE-SH IWV (Fig. 4c) vary from 0.65-1.05 kg m$^{-2}$ throughout the day on 24 December
2018, with an agreement to within 0.1 kg m$^{-2}$ (i.e. ~10-15%), which is consistent with previous
studies (Ricaud et al., 2017).
Observation of clouds from space-borne sensors has two main advantages: 1) it
complements the ground-based cloud observations at Dome C (namely ice/liquid water), and
2) it provides an estimate of the vertical and horizontal extents of the detected cloudy layers.
Note that the CALIPSO spaceborne LIDAR operates at the same wavelength as the backscatter
LIDAR at Dome C, with the same method for discriminating ice from liquid water.
Consequently, the two LIDARs should give consistent information for the detected cloud phase.
However, the presence of an optically thick cloud may extinguish the CALIOP signal
underneath as was already presented in Ricaud et al. (2017) when studying episodes of thick
(5-km deep) clouds and diamond dust (ice crystals in suspension close to the surface). The main
difficulty with this approach is related to the temporal and spatial sampling of the spaceborne
instrument, namely finding a satellite overpass coincident both in time and location with the
cloud observed at Dome C. This, unfortunately, decreases the number of overpasses that is
scientifically exploitable. Nevertheless, on 24 December 2018, 2 orbits of CALIOP/CALIPSO
passed close to Dome C at times when SLW clouds were observed by ground-based
instruments. We show the vertical feature mask and ice/water phase from the pass closest to the
station (~220 km), from 15:50 to 16:03 UTC (Figures 5a and b, respectively). Firstly, we note
the presence of a cloud a few hundred meters deep near the surface in the vicinity of Dome C
(Fig. 5a; note that the CALIOP/CALIPSO altitude is above sea level and Dome C is at an
altitude of 3233 m amsl). Secondly, this cloud is composed of SLW (Fig. 5b), confirming the
analysis based on the observations from the LIDAR and the HAMSTRAD radiometer.
Furthermore, we can state that this SLW cloud is not a local phenomenon but has a horizontal
extent of ~450 km along the orbit track. Considering the CALIOP total and perpendicular
attenuated backscatter data at 532 nm on 24 December 2018 at 16:00 and 14:00 UTC (Figures
Supp4 and Supp5, respectively), we note that: 1) the SLW cloud is located between 3.7 and 3.8
km amsl, that is to say a height from ~450 to ~550 m agl, and 2) since the CALIOP signal is
able to reach the surface underneath the SLW cloud, ice is not detected by the space-borne
instrument. This is consistent with the observations performed at Dome C. The other orbit from
14:11 to 14:25 UTC (Figure Supp6) is slightly more distant than the one shown in Figure 5
(~360 km), but it exhibits a similar SLW cloud located between ~450 and ~550 m agl, over an
even greater horizontal extent of ~700 km along the orbit track.

**4.2. Vertical profiles of temperature and water vapour**
On 24 December 2018, temperatures from both HAMSTRAD and ARPEGE-SH ranged
from 240 to 250 K (-33 to -23°C) from the surface to 1-km agl, compatible with the presence
of SLW clouds. The diurnal variations of temperature and water vapour anomalies calculated
by ARPEGE-SH and measured by HAMSTRAD are shown in Figure 6. For each height, the
daily-averaged value has been subtracted. This has the advantages of highlighting areas of
maximum and minimum changes along the vertical, and reduces biases when comparing the
two data sets. Absolute anomalies (K) are presented for temperatures whilst relative anomalies
(%) are shown for water vapour.
The diurnal variation of the ARPEGE-SH temperature (Fig. 6a) from the surface to 1 km
shows a warm atmosphere before 12:00 UTC and a fast cooling one afterward. HAMSTRAD

shows a similar cooling (Fig. 6b), but the transition is not so abrupt and occurs later, around

15:00 UTC. The diurnal amplitude is greater in ARPEGE-SH (~5 K) than in HAMSTRAD (~3

K). The diurnal variation of the water vapour in ARPEGE-SH (Fig. 6c) from the surface to 1

km shows a wet atmosphere before 12:00 UTC and a drier atmosphere after, again with an

abrupt transition. From HAMSTRAD, the diurnal variation of the water vapour (Fig. 6d) from

the surface to 1 km is more complex, alternating wet and dry phases, which is particularly

obvious at 500-m altitude: wet (00:00-03:00 UTC), dry (03:00-08:00 UTC), wet (08:00-09:00

UTC), dry (09:00-12:00 UTC), wet (12:00-22:00 UTC) and dry (22:00-24:00 UTC). The time

evolution of the SLW cloud (Fig. 2c) and the diurnal variation of the top of the PBL as

calculated by ARPEGE-SH are superposed on all the panels of Figure 6. We note that the SLW

cloud appeared just below the ARPEGE-SH-estimated PBL top, around 08:00 UTC, and

persisted around the same altitude after 12:00 UTC even though the PBL top had dramatically

decreased down to the surface. In addition, the SLW cloud persisted after 12:00 UTC in a layer

that is cooler than earlier in the day, but slightly warmer than the air above and below it.

However, the model shows that this layer is drier while the observations suggest it is wetter.

**4.3. Potential Temperature Gradient**

We now consider the mechanisms that allow the SLW cloud to persist in a thin layer (about

100-m deep) around 500-600 m altitude. Even if the PBL gets thinner after 12:00 UTC, a

residual mixed layer remains above (see e.g. Figure 1.7 of Stull, 2012; Figure 12 top of Ricaud

et al., 2012 and definition of a residual layer from the American Meteorological Society at

http://glossary.ametsoc.org/wiki/Residual_layer). This layer, where turbulence is sporadic or

even absent, lies above the surface-connected stable layer, and can be viewed as a fossil of the

mixed layer developed during the previous mixing period. The transition from the boundary

layer to the free atmosphere is characterized by a local maximum of the potential temperature
($\theta$) vertical gradient ($\partial\theta/\partial z$).
Figure 7 shows $\partial\theta/\partial z$ field and the evolution of the mixed layer top, both computed from
ARPEGE-SH output – the latter defined according to whether the turbulent kinetic energy
exceeds a defined threshold – and the observed SLW cloud superposed. Black areas correspond
to neutral conditions ($\partial\theta/\partial z \sim 0$), whereas the coloured ones relate to stable stratification
according to the colour scale in the Figure. The SLW cloud, once appeared at the top of the
PBL around 08:00 UTC, persists after 12:00 UTC in a layer around 500-600 m coinciding with
the top of the residual mixed layer (see above for the definition) even after the ARPEGE-defined
mixed layer top collapses down to the surface.
Figures 8a, b and c show the vertical profiles of $\theta$ (K) and $\partial\theta/\partial z$ (K km$^{-1}$) as calculated
from temperature measured by the radiosondes and analysed by ARPEGE-SH at Dome C on
24 December 2018 at 00:00 and 12:00 UTC and on 25 December 2018 at 00:00 UTC,
respectively. The presence and the depth of the SLW cloud detected from LIDAR observations
are highlighted in the Figure. The atmosphere as analysed by ARPEGE-SH is about 3-5 K
warmer than the observations. From 100 m upward, the maximum of $\partial\theta/\partial z$ is measured at 400,
550 and 600 m on 24 December 2018 at 00:00 and 12:00 UTC and on 25 December 2018 at
00:00 UTC, respectively with an amplitude of 10, 12 and 40 K km$^{-1}$, respectively. ARPEGE-
SH cannot reproduce the fine vertical structure of $\partial\theta/\partial z$.  For example, the simulated maxima
of $\partial\theta/\partial z$ (Fig. 8) are slightly higher (600, 700 and 600 m for the same dates, respectively) and
less intense than those of radiosondes (8, 8 and 18 K km$^{-1}$, respectively).

**5. Perturbed diurnal cycle of the PBL**
On the second case study, 20 December 2018, the diurnal cycle of the PBL was perturbed
by the sudden arrival of very moist, warm air of oceanic origin. During this warming period,
the boundary layer remains mixed and does not form a stable boundary layer even when the
solar forcing decreases. This will be discussed in detail in the section 7.2.

**5.1. Clouds**

As in section 3.1, the high LIDAR backscatter ($\beta > 100$ $\beta_{mol}$) and low depolarization

(<5%) showed the presence of SLW clouds (Figures 9a, b and c, respectively). Before 13:00
UTC, there is no trace of clouds above Dome C, while from 13:00 to 23:00 UTC SLW clouds
are detected between 200 and 600 m. On all panels, we superimposed the PBL top calculated
by the ARPEGE-SH model. We note that the PBL top does not drop to the surface after 12:00
UTC as typically occurs, like on 24 December 2018, but rather remains between 100 and 200
m. Consistent with the conclusions derived from the observations of 24 December 2018, the
SLW cloud, once present, stays just above the height of the PBL top.

The cloud fraction, ice water and liquid water mixing ratios (kg kg$^{-1}$) calculated by

ARPEGE-SH on 20 December 2018 are shown in Figures 10a, b and c, respectively. Contrary
to the observations, the model simulates mixed-phase clouds (maximum cloud fraction of
~30%), mainly composed of ice, prior to 12:00 UTC; from 00:00 to 06:00 UTC, the clouds are
forecasted below the PBL top. After 12:00 UTC, clouds appear 1-2 hours later in the model
than in the observations, at 14:00-15:00 UTC, just below the PBL top (maximum cloud fraction
of ~100%). The modelled cloud is mainly composed of ice with some traces of SLW above the
PBL around 15:00-16:00 UTC. In all occurrences, the liquid water amounts produced by the
model are extremely small, nearly non-existent. We note the presence of high altitude cirrus
(ice) clouds calculated by ARPEGE-SH after 12:00 UTC around 3-4 km height, while not
observed likely because the LIDAR light is attenuated by the SLW layer. As on 24 December
2018, the model fails to reproduce the presence of the SLW layer observed by the LIDAR near
the PBL top.
The diurnal evolutions of the TCI calculated by ARPEGE-SH, the LWP from
HAMSTRAD and ARPEGE-SH, and the IWV from HAMSTRAD and ARPEGE-SH on 20
December 2018 are presented in Figures 11a, b and c, respectively, with the presence of SLW
clouds derived from the LIDAR observations superimposed on Fig. 11b. Ice clouds are
calculated by ARPEGE-SH mainly around 15:00-16:00 UTC, with TCI values comparable to
those on 24 December 2018. SLW clouds are deduced from HAMSTRAD LWP between 13:00
and 23:00 UTC which coincides well with the SLW clouds observed by the LIDAR. The
maximum LWP values observed during this episode are much higher ($\sim$50 g m$^{-2}$) than on 24
December 2018 ($\sim$2-3 g m$^{-2}$). Again, the ARPEGE-SH LWP is negligible ($\sim$10$^3$ times less than
observations). In parallel with the rapid increase of LWP, the observed IWV also jumps from
$\sim$0.5 to $\sim$2.3 kg m$^{-2}$ within one hour after 13:00 UTC. ARPEGE-SH also calculates an increase
of IWV but lagged by one hour and much less intense ($\sim$1.3 kg m$^{-2}$). Additionally, the model
produces a systematically dryer atmosphere compared to HAMSTRAD by about 0.5 kg m$^{-2}$
after 16:00 UTC, although before the cloudy period that starts at 12:00 UTC, ARPEGE-SH and
HAMSTRAD IWV are consistent to within $\pm$0.2 kg m$^{-2}$.
On 20 December 2018, after 13:00 UTC when SLW clouds have been detected at Dome
C, both CALIPSO overpasses are far away from Dome C and, for the closest overpass at 13:17
UTC (closest distance to Dome C is 500 km), a very thick ice cloud at about 3 km agl prevents
the LIDAR radiation from reaching the surface (Figure Supp7). Unfortunately, no meaningful
information can be ascertained from the spaceborne observations on that day relevant to SLW
clouds in the vicinity of Dome C.

**5.2. Vertical profiles of temperature and water vapour**
The diurnal variations of the temperature and water vapour anomalies on 20 December
2018 as calculated by ARPEGE-SH and measured by HAMSTRAD are shown in Figure 12. In
ARPEGE-SH, a sharp transition between a warm and a cool atmosphere is evident at 12:00
UTC below the top of the PBL. In HAMSTRAD, from 00:00 to 06:00 UTC, the atmosphere
starts warming and then from 06:00 to 13:00 UTC, cools gradually to a minimum. After 13:00
UTC, HAMSTRAD temperatures reveal a warming starting from the surface and progressively
thickening until reaching the top of the PBL by the end of the day. Above the PBL, the
HAMSTRAD-observed and ARPEGE-SH-calculated temporal evolution of temperature and
water vapour are in an overall agreement. In the PBL, the model simulates a moistening around
05:00 UTC, but the most striking event is a sudden drying at 12:00 UTC. In HAMSTRAD,
there is a continuous drying from 00:00 UTC, followed by an obvious transition at 13:00 UTC,
opposite to that of ARPEGE-SH at 12:00 UTC. The warm and wet atmosphere observed after
13:00 UTC develops a mixed layer, consequently the PBL top no longer collapses to a stable
layer, in contrast to what was observed on 24 December. Furthermore, the SLW clouds present
in the entrainment zone steadily remain at the PBL top until the end of the day.

**5.3. Potential Temperature Gradient**
Figure 13 shows $\partial\theta/\partial z$ (K km$^{-1}$) from ARPEGE-SH, with the evolution of the PBL top and
the SLW cloud superimposed. In these perturbed conditions, the SLW clouds are present a few
tens of meters above the top of the PBL after 12:00 UTC. The PBL top is located in a layer
coinciding with the local maximum of $\partial\theta/\partial z$, around 100-300 m, and does not dramatically
decrease to the surface for the rest of the day.
Figures 14a, b and c show the vertical profiles of $\theta$ (K) and $\partial\theta/\partial z$ (K km$^{-1}$) as calculated
from temperature measured by the radiosondes and analysed by ARPEGE-SH at Dome C on
20 December 2018 at 00:00 and 12:00 UTC and on 21 December 2018 at 00:00 UTC,
respectively. The presence and the depth of the SLW cloud detected from LIDAR observations
are highlighted in the Figure. The ARPEGE-SH profiles are about 0-5 K warmer than the
observations. From 50 m upward, the maximum of $\partial\theta/\partial z$ is measured at 75, 150 and 375 m on
20 December 2018 at 00:00 and 12:00 UTC and on 21 December 2018 at 00:00 UTC,
respectively, with a corresponding amplitude of 75, 40 and 55 K km$^{-1}$. The location of the
observed maximum in the potential temperature gradient is consistent with the ARPEGE-SH
calculations on 20 December 2018 prior to the warm and wet episode: at 00:00 UTC (Fig. 14a),
the calculated $\partial\theta/\partial z$ is maximum at 75 m and reaches 100 K km$^{-1}$. However, at 12:00 UTC (Fig.
14b) the modelled $\partial\theta/\partial z$ peaks at 200 m (slightly higher than observed) with a value of 50 K
km$^{-1}$. On the following day at 00:00 UTC (Fig. 14c), $\partial\theta/\partial z$ calculated by ARPEGE-SH shows
two maxima at 100 and 450 m with an amplitude of 45 and 25 K km$^{-1}$, respectively, while the
observations demonstrate a single maximum just below 400 m.

## 569 6. Impact of SLW Clouds on Net Surface Radiation

The presence of clouds over Dome C has a strong impact on the net surface radiation as
demonstrated by Ricaud et al. (2017). This can be seen clearly in the time-series of upwelling
and downwelling longwave and shortwave fluxes observed by BSRN for the two case studies.

### 574 6.1 Typical PBL Case – 24 December 2018

Figure 15 (top) shows the time evolution of the net surface radiation as measured by the
BSRN instruments and as calculated by ARPEGE-SH on 24 December 2018, superimposed
with SLW cloud height. We also show the time evolution of the difference between surface
radiation (W m$^{-2}$) observed by BSRN and calculated by ARPEGE-SH on 24 December 2018,
in longwave downward (LW$\downarrow$), longwave upward (LW$\uparrow$), shortwave downward (SW$\downarrow$) and
shortwave upward (SW$\uparrow$) components, superimposed with LWP. We highlight 4 periods with
images taken from the webcam installed on the shelter hosting the LIDAR and HAMSTRAD:
a) at 00:25 UTC (cirrus clouds, no SLW cloud), b) at 03:56 UTC (cirrus clouds, no SLW cloud),
c) at 09:46 UTC (SLW cloud) and d) at 17:20 UTC (SLW cloud). The net surface radiation
shows maxima between 00:00 and 05:00 UTC (08:00-13:00 LT) and minima between 11:00
and 13:00 UTC (19:00-21:00 LT) in the ARPEGE-SH and BSRN time series. When SLW
clouds are present in the observations (08:00-10:00, 12:00-19:00 and around 21:00 UTC),
whilst absent in ARPEGE-SH, the measured net surface radiation is systematically greater than
the simulated one by 20-30 W m$^{-2}$. In the presence of SLW clouds after 12:00 UTC, this
difference is mainly attributable to LW$\downarrow$ component, BSRN values being 50 W m$^{-2}$ greater than
those of ARPEGE-SH. Thus, SLW clouds tend to radiate more LW radiation toward the ground
(like greenhouse gases) than more transparent clouds, like cirrus, do. There are differences from
-30 to +60 W m$^{-2}$ between observed and calculated SW$\downarrow$ and SW$\uparrow$ components but this
difference falls within $\pm$10 W m$^{-2}$ for the net SW surface radiation (SW$\downarrow$ - SW$\uparrow$). The reflective
impact of SLW layers can also be seen after 12:00 UTC: unlike observed SLW clouds,
ARPEGE-SH simulates ice clouds, and therefore too high SW$\downarrow$ values. The difference between
observed and simulated values of this parameter thus increases, as can be seen on the Figure.
But because of the high values in surface albedo, a compensating effect occurs on the surface
reflected SW fluxes, and the resulting impact on net radiation is quite weak (the time series of
the observed – simulated difference in incoming and reflected SW flux follow each other quite
well). The major impact on net radiation is therefore related to the longwave fluxes.

**6.2 Perturbed PBL Case – 20 December 2018**
Figure 16 (top) shows the net surface radiation as measured by the BSRN photometric
instruments and as calculated by ARPEGE-SH for 20 December 2018, superimposed with the
SLW clouds. We also show the time evolution of difference in surface radiation (W m$^{-2}$)
observed by BSRN and calculated by ARPEGE-SH on 20 December 2018 for LW$\downarrow$, LW$\uparrow$,
SW$\downarrow$ and SW$\uparrow$ components, superimposed with LWP. We highlight 4 periods with snapshots

taken from the webcam: 1) 07:15 UTC (clear sky), 2) 12:35 UTC (clear sky), 3) 13:30 UTC (SLW cloud) and 4) 21:00 UTC (SLW cloud). Before 13:00 UTC, there are no clouds above Dome C whilst after 13:00 UTC clouds are present. The diurnal evolution of the modelled and observed net surface radiation shows a maximum of ~+50 W m$^{-2}$ in ARPEGE-SH and ~+85 W m$^{-2}$ in BSRN over the period 00:00-04:00 UTC, and a minimum of about -50 W m$^{-2}$ around 12:00-13:00 UTC on both time series. Nevertheless, when SLW clouds are observed at 13:00 UTC, the observed net surface radiation jumps to +10 W m$^{-2}$, a feature not reproduced in the model. The difference between the BSRN-observed and ARPEGE-SH-modelled net surface radiation is larger than +30 W m$^{-2}$ when SLW clouds are present, reaching +60 W m$^{-2}$ when the LWP measured by HAMSTRAD is at its maximum (50 g m$^{-2}$ at 13:00 UTC). This is twice the difference observed in the non-perturbed PBL episode detailed in section 3.4. This underlines again the strong impact SLW clouds may have on the radiation budget over Antarctica. In the presence of SLW clouds after 13:00 UTC, the difference in net surface radiation is mainly attributable to LW↓ component, BSRN values being 100 W m$^{-2}$ greater than those of ARPEGE-SH. The SW↓ and SW↑ also decrease due to the high reflectivity of the SLW layer seen at 12:00 UTC and again at 15:00 UTC. Note that there are differences from -100 to +60 W m$^{-2}$ between observed and calculated SW↓ and SW↑ components but this difference falls below 20 W m$^{-2}$ for the net SW surface radiation (SW↓ - SW↑). Both SW components decrease after 17:00 UTC. Some of this may be due to: 1) increasing LWP, and 2) the presence of precipitating ice crystals and/or blowing snow (characterized by red spots on Figure 9b) that are increasing optical depth and decreasing transmission/visibility (webcam images in Figure 16d) although surface wind was rather weak (3-10 m s$^{-1}$, not shown).

## 7. Discussions

### 7.1. SLW Clouds vs Mixed-Phase Clouds

In order to evaluate whether the observed cloud is constituted of liquid and/or mixed phase
water, we have considered the raw signals recorded by the LIDAR. For the two dates under
consideration (Figures Supp8 and Supp9 relative to 24 and 20 December 2018, respectively),
we have represented (top) the P signal as the signal received with the same polarization as the
laser (unpolarized component). Any suspended object can contribute to P signal. We have also
represented the S (cross-polarized) LIDAR signal (bottom) that is only produced by non-
spherical (obviously frozen at Dome C) particles and, to a smaller extent, by multiple scattering
in water clouds.
First of all, an elevated P signal above ~400 m on 24 December 2018 (P ≥ 0.1 mV) and
above ~200 m on 20 December 2018 (P ≥ 0.3 mV) is associated with a cloud as shown in
sections 4.1 and 5.1. Inside these clouds, the S signal is always very low: S ~0.003 mV on 24
December 2018 and ~0.01 mV on 20 December 2018. Consequently, the S signal is very weak
and corresponds to a maximum of ~3% of the corresponding P signal. Some S signal is
nevertheless present in the cloud and could be given by multiple scattering inside the truly liquid
water cloud and/or the effective presence of ice particles.
When considering the LIDAR depolarization diurnal evolutions presented in Figures 2b
and 9b associated to the two dates, ice particles could have been disappeared in the low
depolarization ratio S/P of the SLW layer because the P signal inside the SLW cloud is very
high compared to the S signal. But when considering the P and S signals distinctively (Figs.
Supp8 and Supp9), the S signal remains very weak in the SLW cloud compared to the P signal
whatever the date considered. Consequently, even if the presence of some ice particles scattered
within the SLW layers cannot be excluded from the S signal plot, the very low depolarization
of the layers leads to classify them as a liquid cloud.
The important point is that the optical properties of the layer, relevant for the radiative
budget in the shortwave, such as optical extinction, optical depth, asymmetry factors, etc. are
bound to the P signal, being e.g. optical extinction in the visible proportional to the lidar P
signal. Thus, the shortwave radiative characteristics of the cloud are driven by the P signal, and
thus by liquid water.
On the other hand, when we consider the aerosol depolarization ratio measured by the
LIDAR (Figure 2b) and the total snow flux calculated by ARPEGE-SH (Figure Supp2) on 24
December 2018, it is obvious that solid precipitation is present from 00:00 to 10:00 UTC in a
layer from ~500 m to the surface (vertical stripes). Therefore, physical processes are occurring
within the cloud to deplete liquid and turn it into solid, causing the ice observed and calculated
below the SLW layer. In this case, the ice microphysics would also be important since it leads
to the termination of the SLW layer, hence indirectly impacting the radiative budget. As a
consequence, we cannot completely rule out the possibility that this is a SLW layer of an overall
mixed-phase cloud.

**7.2. SLW Clouds and PBL**
During the YOPP SOP-SH, SLW clouds were observed in the LIDAR data for 15 days in
December (49% of days) and 13 days in January (47%), which is a similar rate of occurrence
to other years (53% in December 2016 and 2018; 51% in January 2018 and 2019) (Figure 17).
A day is flagged with a SLW cloud occurrence when a SLW cloud has been detected in the
LIDAR observations for a period longer than 1 hour. The clouds observed during the SOP-SH
are typically located at the top of the PBL (100 to 400 m height) and are 50-100 m thick.
The presence of SLW clouds in the atmosphere is strongly dependent on the temperature
field. From Fig. 2.33 of Pruppacher and Klett (2012), the percentage of clouds containing no
ice becomes non-negligible at temperatures greater than -35°C, although SLW clouds have been
observed at lower temperatures over Russia (-36°C) and the Rocky Mountains in the USA (-
40.7°C). Recent laboratory measurements show that liquid water can exist down to -42.55°C
(Goy et al., 2018).

Considering that the SLW clouds at Dome C are so thin, they resemble stratocumulus, as

can be observed at middle latitudes. The diurnal cycle of the SLW cloud also evokes that of
oceanic stratocumulus, with a trend to fragmentation and/or dissipation during the "day" (local
noon) because of solar absorption and to a solid deck state during the "night" (local midnight)
because of reversed buoyancy due to cloud top longwave cooling. We use here the "night" and
"day" terms for convenience, though solar radiation remains positive 24-hr long at this period
of the year. During the SOP-SH, SLW clouds were observed in the LIDAR data for
approximately 48% of days (Fig. 17) but it is not yet evident whether they were formed during
the "day" (local noon) when the mixed layer becomes thick enough to reach the condensation
level, and vertically broadened during the "night", or created during the "night" (local midnight)
and then dissipated during the coming "day". Complementary observations would be needed,
in particular turbulence profiles from the surface to above the top of boundary-layer clouds, to
determine what is the coupling/decoupling diurnal cycle of these clouds.

The diurnal evolution of the top of the PBL is consistent with previous studies carried out

at Dome C (e.g. Argentini et al., 2005; King et al., 2006; Ricaud et al., 2012; Casasanta et al.,
2014), with a top higher when there is a relatively warm mixed layer than in colder stable
conditions.

The colocation of the positive potential temperature gradient with the height of the SLW

clouds is consistent with the schematic representation of the diurnal variation of the PBL
illustrated by Stull (2012) and adapted by Ricaud et al. (2012) for the Eastern Antarctic Plateau.
Figure 18 is a modified version of Figure 12 from Ricaud et al. (2012) to take into account the
impact of the clouds on the PBL structure. Starting with the simplest, cloud-free case, we have
during the convective (mixing) period a mixed layer at the top of which is located the
"entrainment zone", so-named because air parcels coming from the above free troposphere are
entrained into the mixed layer below under the effect of overshooting thermals and
compensating descending currents. When clouds form at the top of the PBL (boundary-layer
clouds), we consider that the PBL locally (i.e. where clouds are present) extends to the top of
these clouds. The PBL is clearly separated from the above stable free troposphere by the so-
called "capping inversion". The cloud layers as well as the capping inversion zone are thin, of
the order of 100 m. When the stable layer forms close to the surface, the SLW cloud may persist
over the residual mixed layer, as may persist the capping inversion zone which can also be
qualified as "residual". The stable layer is then progressively eroded, when the incoming
available energy becomes large enough to ensure turbulent mixing from the surface. The new
mixing layer thus grows through the previous stable layer and residual mixed layer, up to it
reaches the residual capping inversion. The stratification of the different layers is characterized
by the simplified potential temperature profiles in Figure 18. Considering both the potential
temperature gradients and the vertical extent of the SLW cloud, these layers are quite thin, less
than 100-m deep.

**7.3 SLW Clouds in ARPEGE-SH**
In comparison with observations, ARPEGE-SH consistently underestimates LWP by
several orders of magnitude. This is due in part to the partitioning into liquid and ice phases in
the model which is a simple function of temperature such that, below -20°C, all cloud particles
are iced. The inability of ARPEGE-SH to reproduce the observed liquid water content of the
cloud leads to an underestimate of the simulated downwelling longwave radiation relative to
observations, and an overestimate of both upwelling and downwelling shortwave flux. This
effect is particularly notable in the perturbed PBL case study where the high moisture content
leads to an enhanced longwave effect. As the SLW cloud horizontal extent in the first case
study is between ~450 and ~700 km and persists over more than 12 hours (section 4.1), the
discrepancy in the net surface radiation between observation and NWP model may have a strong
impact on the calculation of the radiation budget over Antarctica. Lawson and Gettelman (2014)
showed that better representation of liquid water in modelled mixed phase clouds in Global
Climate Models led to an increase of 7.4 W m$^{-2}$ in the cloud radiative effect over Antarctica.
In Figure 17, we show the percentage of days per month that SLW clouds were detected
within the LIDAR data for more than 12 hours per day (blue) during SOP-SH. As expected,
SLW clouds with a minimum duration of 12 hours (blue) occur less often than SLW clouds
with a minimum duration of 1 hour (green). But, whatever the criterium used (1 hour or 12
hours), the maxima of SLW cloud presence occur in December and January during SOP-SH.
12-h SLW clouds occurred about a quarter of the days (20-25%) compared to roughly half of
the days for 1-h SLW clouds (40-45%). This reinforces the argument of the critical importance
of well representing SLW clouds in models in order to better estimate radiation budget over
Antarctica.
Furthermore, even when considering analyses of ARPEGE-SH at 00:00, 06:00, 12:00 and
18:00 UTC and associated forecasts (not shown), neither IVW nor LWP are significantly
modified, and SLW remains underestimated. The 4Dvar analysis is not able to correct the dry
bias especially during the case of 20 December 2018 probably because it is influenced by a
large-scale advection. The underestimation of the SLW in ARPEGE-SH can be explained by
the fact that: 1) the underestimation of liquid water is mainly a physical problem in the model
related to the ice/liquid partition function vs temperature (see below) and 2), since the cloud
water is not a model control variable in the 4DVar scheme, it cannot be updated by the analysis
step of the 4DVar data assimilation process.
We have thus tried to modify the ice partition function (ice/liquid water vs temperature)
used in the ARPEGE-SH operational model (Figure Supp10). We noticed that, for temperatures
below -20°C, water was present only in the solid form in the model. A test has been performed
for 20 and 24 December 2018 with ARPEGE-SH by considering a new ice partition function
allowing the presence of liquid water for temperature between -20°C and -40°C (Figure
Supp10). The analyses were done at 00:00 UTC and the forecasts from 01:00 to 24:00 UTC.
This run was labelled as ARPEGE-SH-TEST.

For 24 December 2018, and consistently with Fig. 3, we have drawn on Fig. Supp11 the

diurnal evolutions of different variables calculated by ARPEGE-SH-TEST: a) the Cloud
Fraction, b) the Ice Water mixing ratio and c) the Liquid Water mixing ratio. Similarly, and
consistently with Fig. 4, Figure 19 presents: a) the ARPEGE-SH-TEST TCI, b) the LWP
measured by HAMSTRAD and calculated by ARPEGE-SH-TEST and c) the IWV measured
by HAMSTRAD and calculated by ARPEGE-SH-TEST. Eventually, and consistently with Fig.
9, Figure Supp13 presents the net surface radiation observed by BSRN and calculated by
ARPEGE-SH-TEST, and the difference between surface radiation of longwave downward,
longwave upward, shortwave downward and shortwave upward components observed by
BSRN and calculated by ARPEGE-SH-TEST. In the same manner, for the case of 20 December
2018, Figs. Supp12, 20 and Supp14 echo Figs. 11, 12 and 16, respectively.

On 24 December 2018 (typical case), the new partition function significantly improves the

modelled SLW, with liquid water content about 1000 times greater in ARPEGE-SH-TEST than
in ARPEGE-SH, and LWP varying from ~0 to ~3 g m$^{-2}$ consistently with HAMSTRAD to
within ±0.5 g m$^{-2}$. The impact on the net surface radiation is obvious with an excellent
agreement between ARPEGE-SH-TEST and BSRN to within ±20 W m$^{-2}$. Unfortunately, on
20 December 2018 (perturbed case), even if the impact on SWL clouds is important (liquid
water content multiplied by a factor 100), LWP is still a factor 10 less in ARPEGE-SH-TEST
than in HAMSTRAD. ARPEGE-SH-TEST still fails to reproduce the large increase in liquid
water and IWV at 13:00 UTC since the local maximum is calculated 2 hours later. The impact
on the net surface radiation is weak with ARPEGE-SH-TEST underestimating the net surface
radiation by 50 W m$^{-2}$ compared to observations, mainly attributable to the downwelling
longwave surface radiation from BSRN being 100 W m$^{-2}$ greater than that of ARPEGE-SH-
TEST.
Finally, the bias on the net surface radiation and the underestimation of IWV and LWP of
the model compared to the observations is strongly reduced when using a new ice partition
function in ARPEGE-SH-TEST. This suggests that LWP has more impact than IWV on LW↓
due to the small quantities of specific humidity at Dome C.

**8. Conclusions**

A comprehensive water budget study has been performed during the Year of Polar Programs
SOP-SH at Dome C (Concordia, Antarctica) from mid-November 2018 to mid-February 2019.
Supercooled liquid water (SLW) clouds were observed and analysed by means of remote-
sensing ground-based instrumentation (tropospheric depolarization LIDAR, HAMSTRAD
microwave radiometer, BSRN net surface radiation), radiosondes, spaceborne sensor
(CALIOP/CALIPSO depolarization LIDAR) and the NWP ARPEGE-SH. The analysis shows
that SLW clouds were present from November to March, with the greatest frequency occurring
in December and January since ~50% of the days in summer time exhibited SLW clouds for at
least one hour. The clouds observed during the SOP-SH are typically located at the top of the
boundary layer (100 to 400 m height) and are 50-100 m thick.
The analyses focused on two periods showing 1) a typical diurnal cycle of the PBL on 24
December 2018 (warm and dry, local mixing layer followed by a thinner cold and dry, local
stable layer which develops when the surface has cooled down) and 2) a perturbed diurnal cycle
of the PBL on 20 December 2018 (a warm and wet episode prevented from a clear diurnal cycle
of the PBL top). In both cases thin (~100-m thick) SLW clouds have been observed by ground-
based and spaceborne LIDARs developing within the entrainment and the capping inversion
zones at the top of the PBL. Spaceborne LIDAR observations revealed horizontal extensions of
these clouds as large as 700 km for the 24 December case study. ARPEGE-SH was not able to
correctly estimate the ratio between liquid and solid water inside the cloudy layers, with SLW
always strongly underestimated by a factor 1000 in the studied cases, mainly because the
liquid/ice partition function used in the model favours ice at temperatures less than -20°C.
Consequently, the net surface radiation was affected by the presence of SLW clouds during
these two episodes. The net surface radiation observed by BSRN was 20-30 W m$^{-2}$ higher than
that modelled in ARPEGE-SH on 24 December 2018 (typical diurnal cycle of the PBL), this
difference reaching +50 W m$^{-2}$ on 20 December 2018 (perturbed diurnal cycle of the PBL),
consistent with the total observed liquid water being 20 times greater in the perturbed PBL
diurnal cycle than in the typical PBL diurnal cycle. The difference in the net surface radiation
is mainly attributable to longwave downward surface radiation, BSRN values being 50 and 100
W m$^{-2}$ greater than those of ARPEGE-SH in the typical and perturbed cases, respectively.

The ice/liquid partition function used in the ARPEGE-SH NWP has been modified to favour

liquid water at temperatures below -20°C down to -40°C. For the two study cases, the model
run with this new partition function has been able to generate SLW clouds. During the typical
case, modelled LWP was consistent with observations and, consequently, the net surface
radiation calculated by the model agreed with measurements to within ±20 W m$^{-2}$. During the
perturbed case, modelled LWP was a factor 10 less than observations and, consequently, the
model underestimated the net surface radiation by ~50 W m$^{-2}$ compared to observations.

Time coincident ground-based remote-sensed measurements of water (vapour, liquid and

solid), temperature and net surface radiation are available at Dome C since 2015. Consequently,
a comprehensive statistical analysis of the presence of SLW clouds will be performed in the
near future. Coupled with modelling studies (NWP ARPEGE-SH, mesoscale models), an
estimation of the radiative impact of these clouds on the local climate will then be performed.

**Data availability**
HAMSTRAD data are available at http://www.cnrm.meteo.fr/spip.php?article961&lang=en
(last access:  28 August 2019). The CALIOP images are accessible at http://www-
calipso.larc.nasa.gov/ (last access:  28 August 2019). The tropospheric depolarization LIDAR
data are reachable at http://lidarmax.altervista.org/englidar/_Antarctic%20LIDAR.php (last
access:  28 August 2019). Radiosondes are available at http://www.climantartide.it (last access:
28 August 2019). BSRN data can be obtained from the ftp server (https://bsrn.awi.de/data/data-
retrieval-via-ftp/) (last access:  28 August 2019). The ARPEGE data and corresponding
technical information are available from the YOPP Data Portal and from the ftp server (ftp.umr-
cnrm.fr with user: yopp and password: Arpege) (last access:  28 August 2019). The NCEP data
are available at https://www.esrl.noaa.gov/psd/ and the back-trajectory calculations can be
performed at https://www.ready.noaa.gov/HYSPLIT.php.

**Author contributions**
PR, MDG, AL, and PG provided the observational data while EB, NA and VG developed
the model code and performed the simulations. PD, JLA and DV contributed to the data
interpretation. All the co-authors participated in the data analysis. PR prepared the manuscript
with contributions from all co-authors.  DV, EB, NA, MDG and PD also contributed
significantly to the revision of the manuscript supervised by PR.

**Competing interests**
The authors declare that they have no conflict of interest.

**Acknowledgments**

The present research project Water Budget over Dome C (H2O-DC) has been approved by the Year of Polar Prediction (YOPP) international committee. The HAMSTRAD programme (910) was supported by the French Polar Institute, Institut polaire français Paul-Emile Victor (IPEV), the Institut National des Sciences de l'Univers (INSU)/Centre National de la Recherche Scientifique (CNRS), Météo-France and the Centre National d'Etudes Spatiales (CNES). The permanently manned Concordia station is jointly operated by IPEV and the Italian Programma Nazionale Ricerche in Antartide (PNRA). The tropospheric LIDAR operates at Dome C from 2008 within the framework of several Italian national (PNRA) projects. We would like to thank all the winterover personnel who worked at Dome C on the different projects: HAMSTRAD, aerosol LIDAR and BSRN. The authors also acknowledge the CALIPSO science team for providing the CALIOP images. We acknowledge the NCEP_Reanalysis 2 data provided by the NOAA/OAR/ESRL PSD, Boulder, Colorado, USA, from their Web site at https://www.esrl.noaa.gov/psd/ and the NOAA Air Resources Laboratory to have accessed the HYSPLIT model through https://www.ready.noaa.gov/HYSPLIT.php. We would like to thank the two anonymous reviewers for their beneficial comments.

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

**Figures**

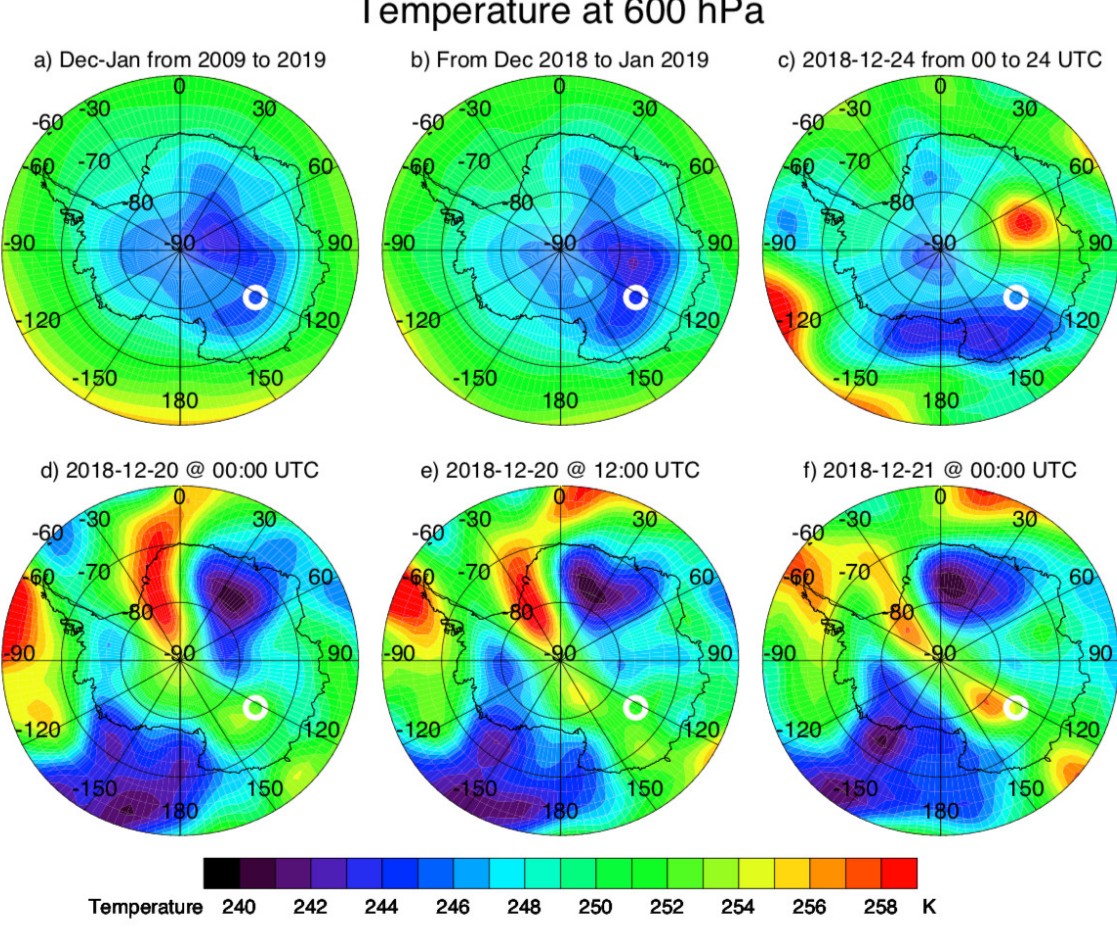

1010

**Figure 1:** Temperature fields from NCEP at 600 hPa: a) decadal average over December-January from 2009 to 2019, b) YOPP average over December 2018-January 2019, c) daily average over 24 December 2018, d) 20 December 2018 at 00:00 UTC, e) 20 December 2018 at 12:00 UTC, and f) 21 December 2018 at 00:00 UTC. The white circle represents the position of the Dome C station.



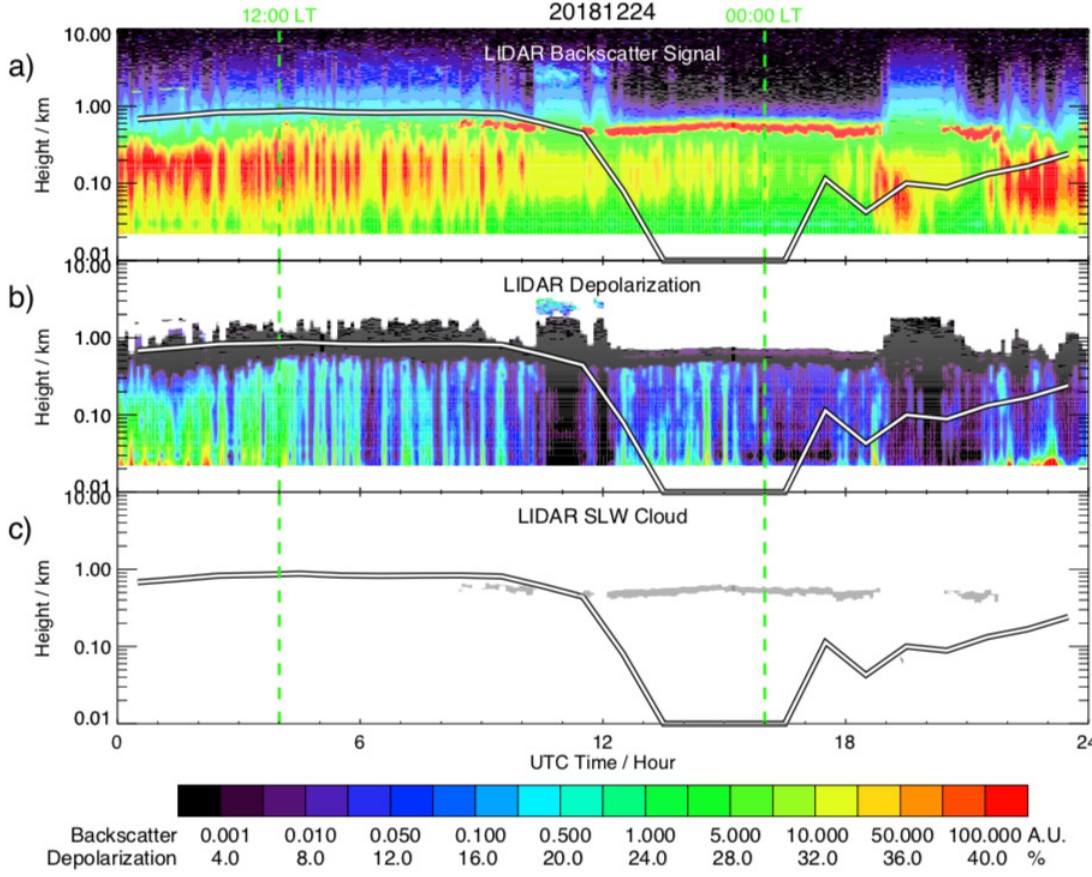


**Figure 2:** Diurnal variation on 24 December 2018 (UTC Time) along the vertical of: a) the
backscatter signal (Arbitrary Unit, A.U.), b) the depolarization ratio (%) measured by the
aerosol LIDAR, and c) the Supercooled Liquid Water (SLW) cloud height (grey) deduced from
the aerosol LIDAR ($\beta_c > 100\ \beta_{mol}$, depolarization < 5%). Superimposed to all the Figures is the
top of the Planetary Boundary Layer calculated by the ARPEGE-SH model (black-white thick
line). Two vertical green dashed lines indicate 12:00 and 00:00 LT.


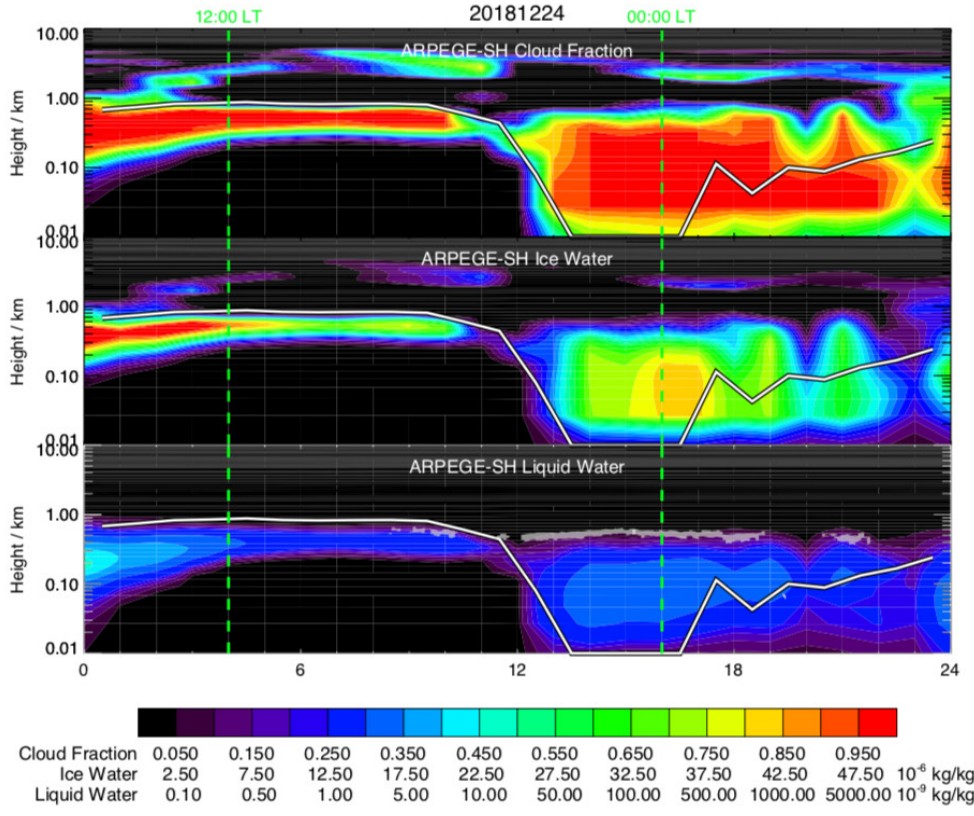


**Figure 3:** Time-height cross section on 24 December 2018 (UTC Time) of: a) the Cloud
Fraction (0-1), b) the Ice Water mixing ratio ($10^{-6}$ kg kg$^{-1}$) and c) the Liquid Water mixing ratio
($10^{-9}$ kg kg$^{-1}$) calculated by the ARPEGE-SH model. Superimposed to all the panels is the top
of the Planetary Boundary Layer calculated by the ARPEGE-SH model (black-white thick line).
Superimposed in panel c is the SLW cloud (grey area) deduced from the LIDAR observations
(see Fig. 1c). Two vertical green dashed lines indicate 12:00 and 00:00 LT.


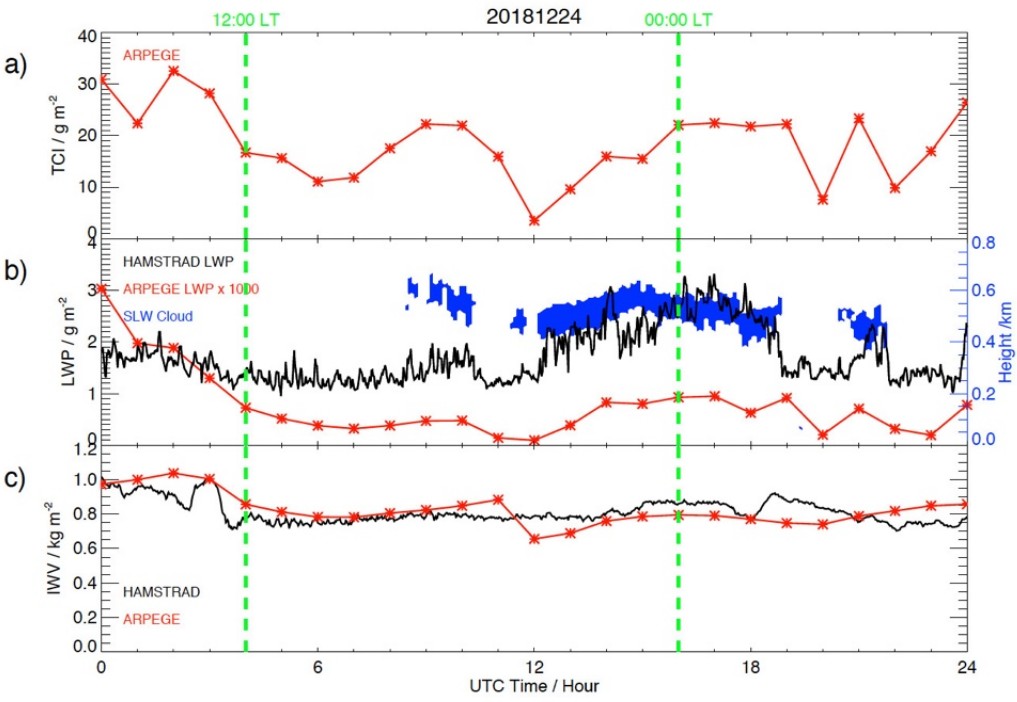


**Figure 4:** Diurnal variation on 24 December 2018 (UTC Time) of: a) the Total Column of Ice
(TCI) (g m$^{-2}$) calculated by ARPEGE-SH (red crossed line), b) the Liquid Water Path (LWP)
measured by HAMSTRAD (g m$^{-2}$, black solid line) and calculated by ARPEGE-SH (x1000 g
m$^{-2}$, red crossed line) and c) the Integrated Water Vapour (IWV, kg m$^{-2}$) measured by
HAMSTRAD (black solid line) and calculated by ARPEGE-SH (red crossed line).
Superimposed to panel b) is the SLW cloud thickness (blue area) deduced from the LIDAR
observations (see Fig. 1c) (blue y-axis on the right of the Figure). Note LWP from ARPEGE-
SH has been multiplied by a factor 1000. Two vertical green dashed lines indicate 12:00 and
00:00 LT.


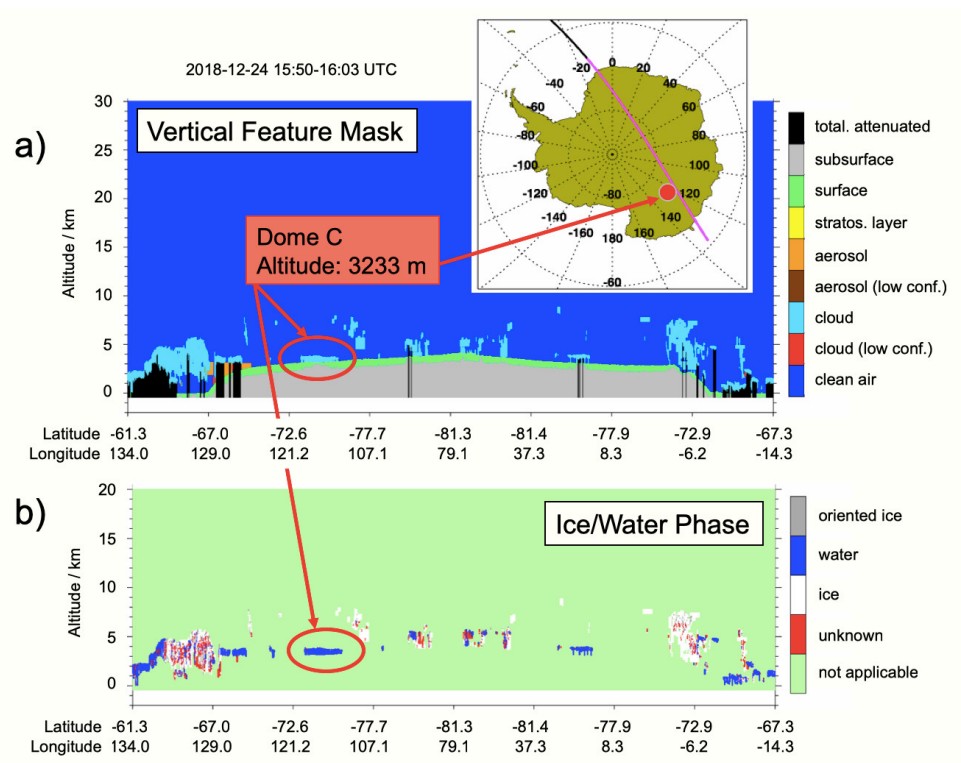

Figure 5: CALIOP/CALIPSO spaceborne LIDAR observations version V3.40 along one orbit on 24 December 2018 (15:50-16:03 UTC) in the vicinity of Dome C (75°S, 123°E): a) the Vertical Feature Mask highlighting a cloud (light blue) near the surface (red circle) and b) the Ice/Water Phase Mask highlighting a SLW (dark blue) cloud near the surface (red circle). The ground-track of the sensor (pink) has been embedded at the top of the Figure, with the location of Dome C marked (red filled circle). Note that the altitude is relative to the sea surface, with the height of surface of Dome C at an elevation of 3233 m amsl. Figure adapted from the original image available at https://www-calipso.larc.nasa.gov/products/lidar/browse_images/std_v34x_showdate.php?browse_date=2018-12-24.

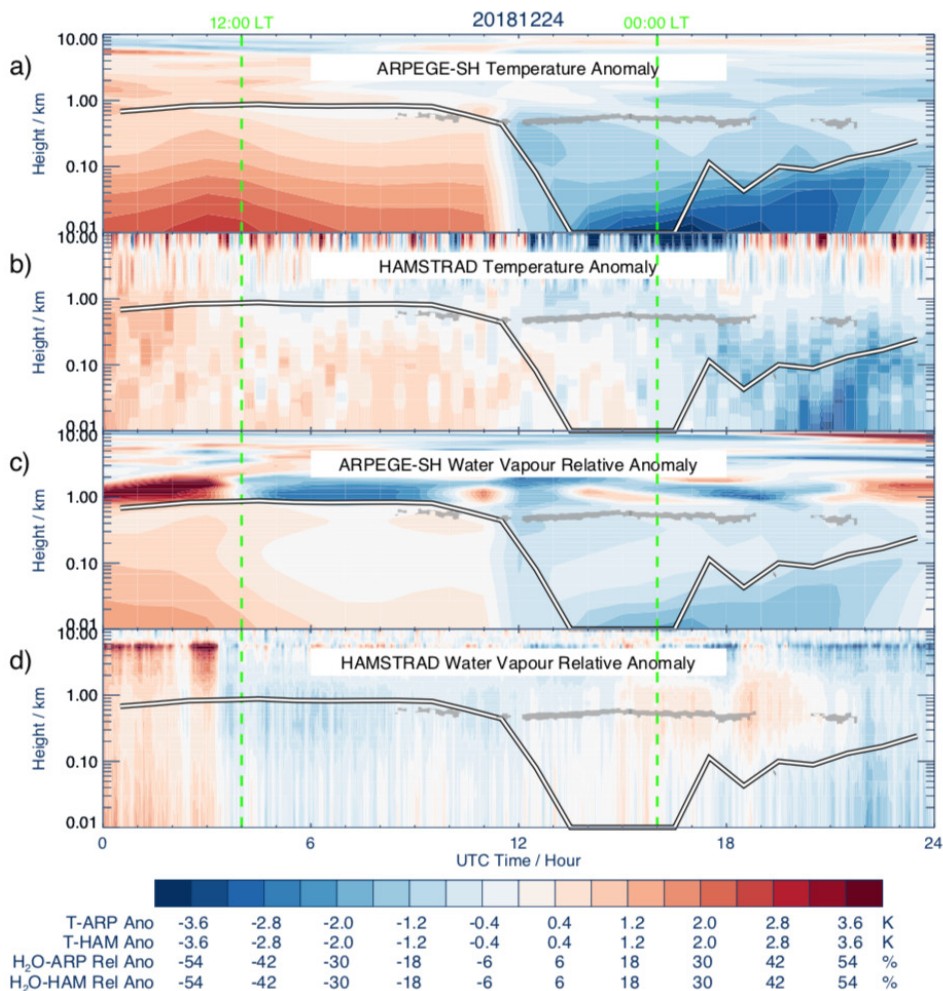

**Figure 6:** Time-height cross section on 24 December 2018 (UTC Time) of a) the temperature anomaly (K) calculated by ARPEGE-SH and b) observed by HAMSTRAD, c) the water vapour relative anomaly (%) calculated by ARPEGE-SH and d) observed by HAMSTRAD. Superimposed to all the Figures are the SLW cloud altitude (grey area) deduced from the LIDAR observations (see Fig. 1c) and the top of the Planetary Boundary Layer calculated by the ARPEGE-SH model (black-white thick line). Two vertical green dashed lines indicate 12:00 and 00:00 LT.

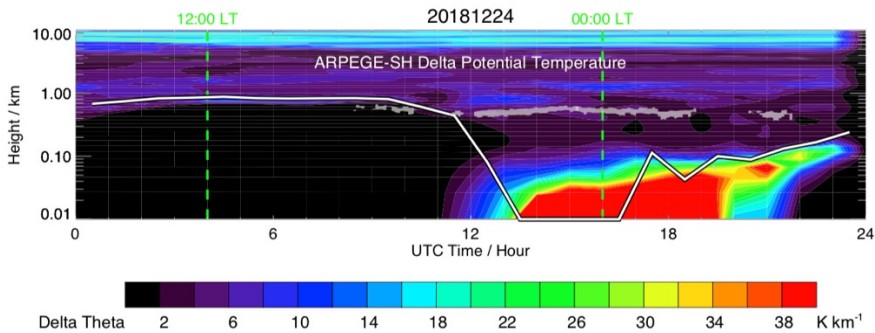


**Figure 7:** Time-height cross section of $\partial\theta/\partial z$ (K km$^{-1}$) calculated from ARPEGE-SH
temperature on 24 December 2018 (UTC Time). Superimposed are the SLW cloud altitude
(grey area) deduced from the LIDAR observations (see Fig. 1) and the top of the Planetary
Boundary Layer calculated by the ARPEGE-SH model (black-white thick line). Two vertical
green dashed lines indicate 12:00 and 00:00 LT.



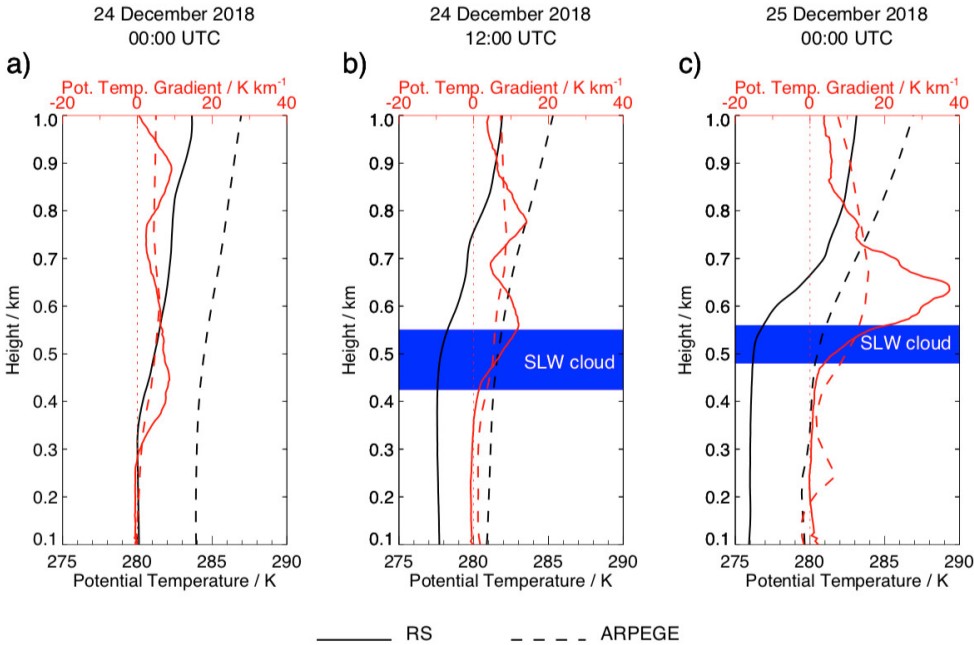


**Figure 8:** Vertical profiles of potential temperature θ (black) and the gradient in potential
temperature ∂θ/∂z (red) as calculated from temperature measured by the radiosondes (solid line)
and analysed by ARPEGE-SH (dashed line) at Dome C on 24 December 2018 at a) 00:00 and
b) 12:00 UTC, and c) on 25 December 2018 at 00:00 UTC. The presence and the depth of the
SLW cloud detected from LIDAR observations are indicated by a blue area.

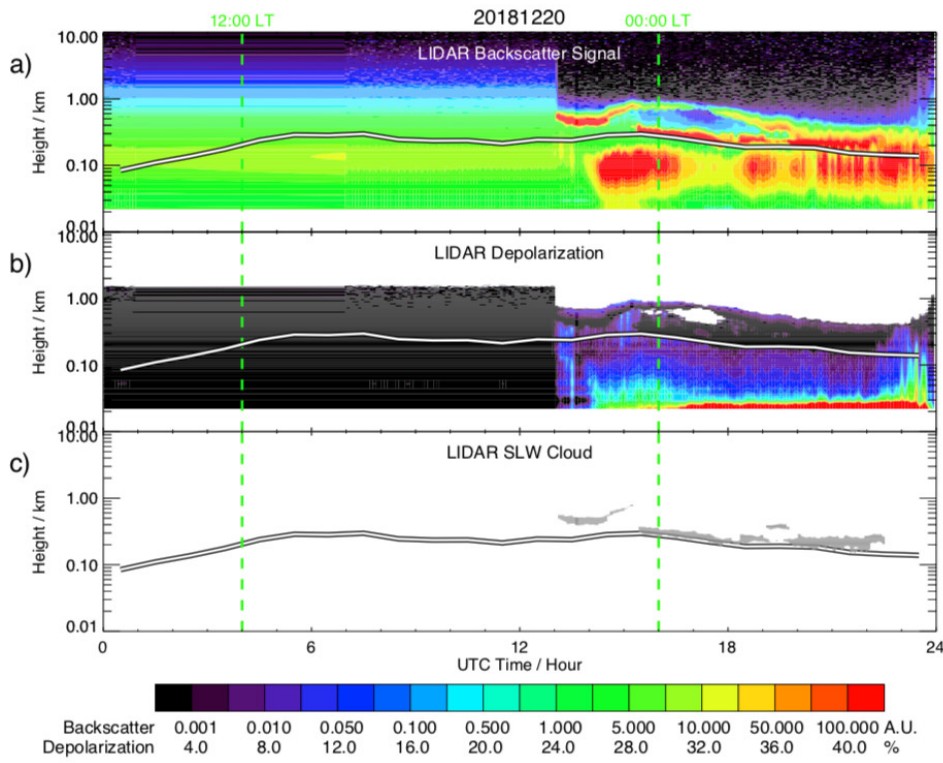


**Figure 9:** Same as Figure 2 but for 20 December 2018.

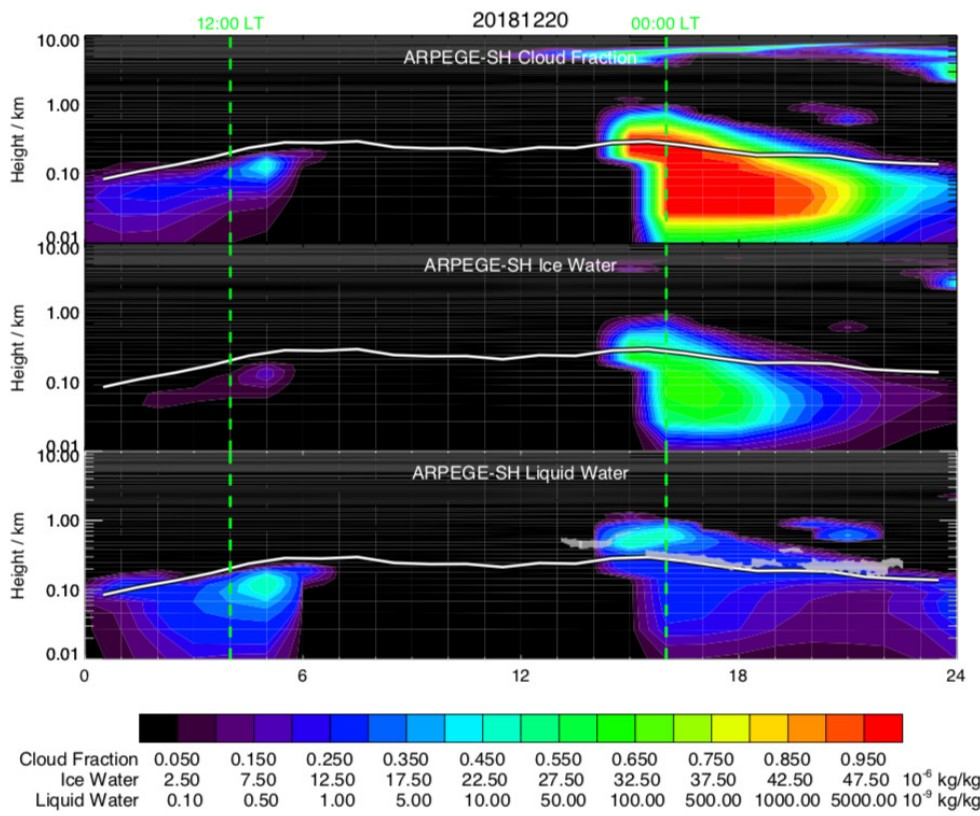


**Figure 10:** Same as Figure 3 but for 20 December 2018.

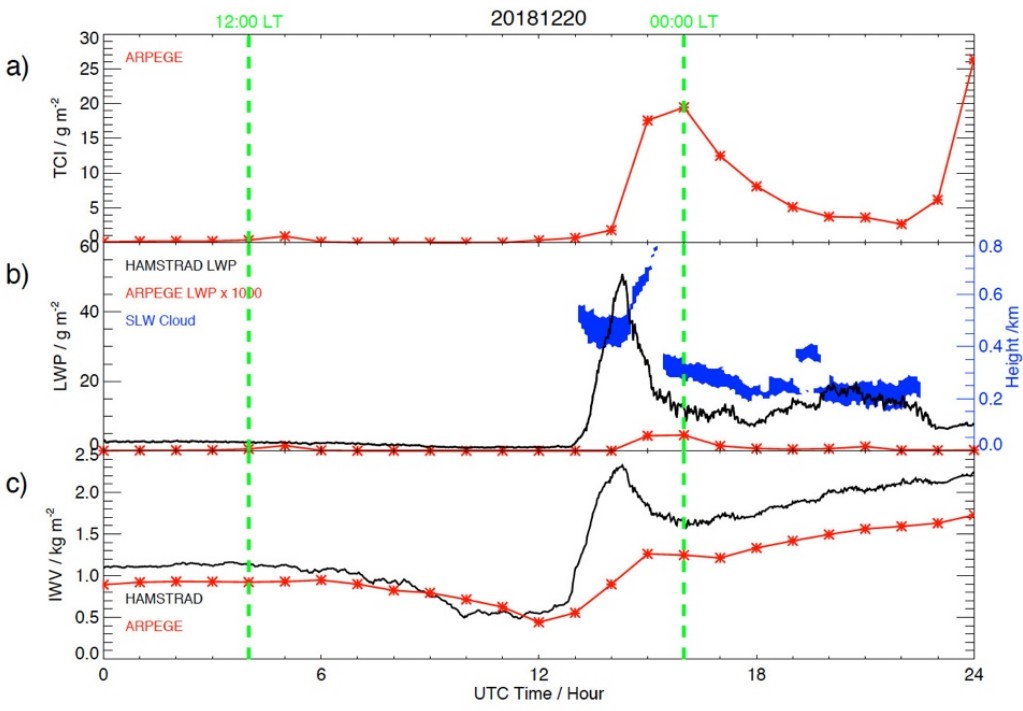


**Figure 11:** Same as Figure 4 but for 20 December 2018.

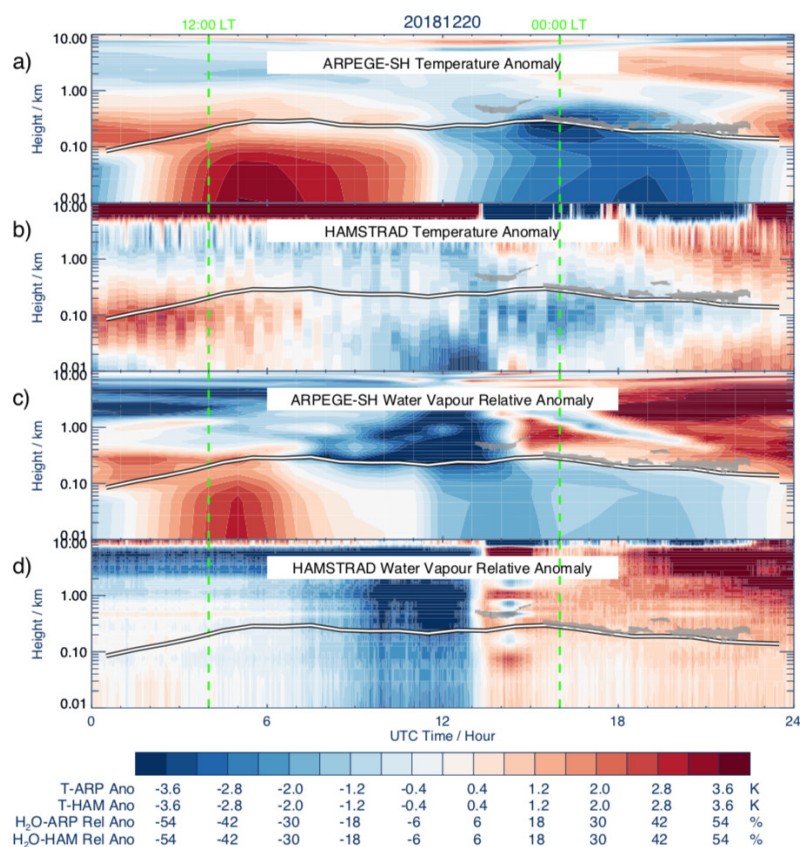


**Figure 12:** Same as Figure 6 but for 20 December 2018.

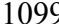

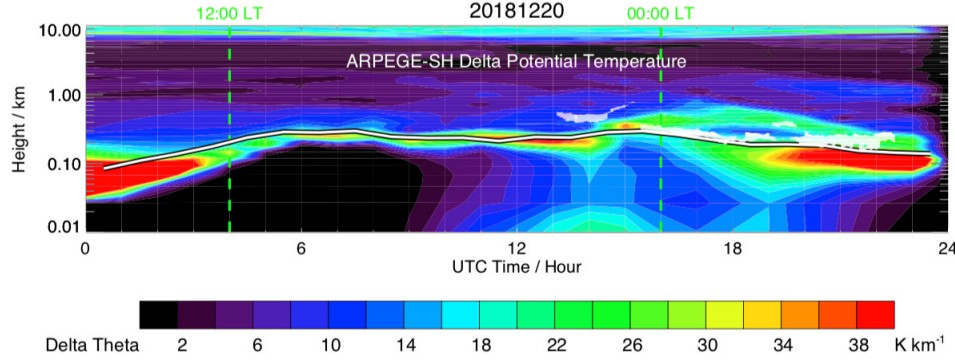


**Figure 13:** Same as Figure 7 but for 20 December 2018.


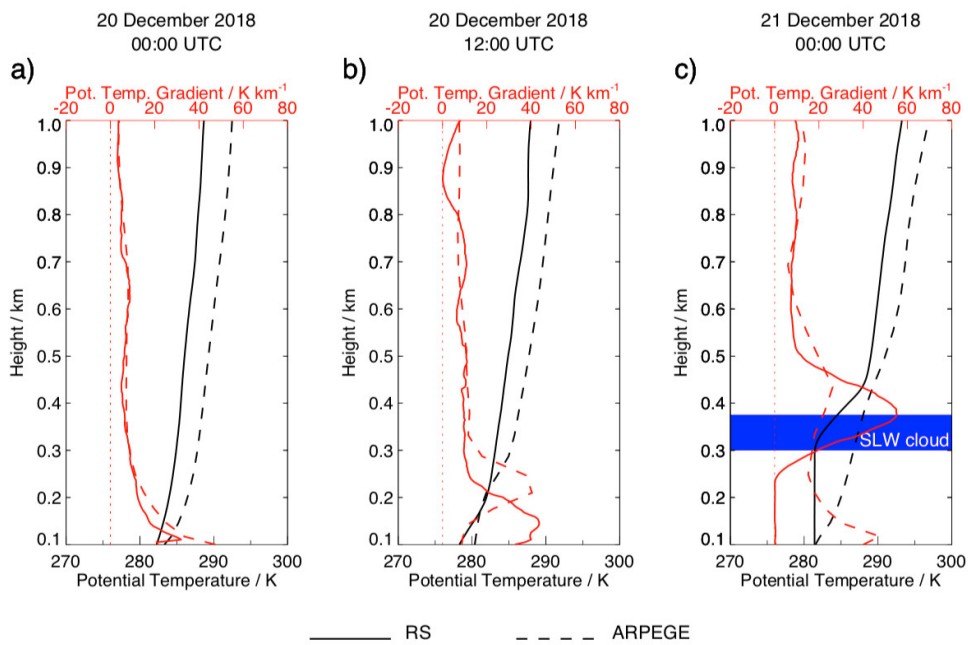


**Figure 14:** Same as Figure 8 but on 20 December 2018 at a) 00:00 and b) 12:00 UTC, and c)
on 21 December 2018 at 00:00 UTC.

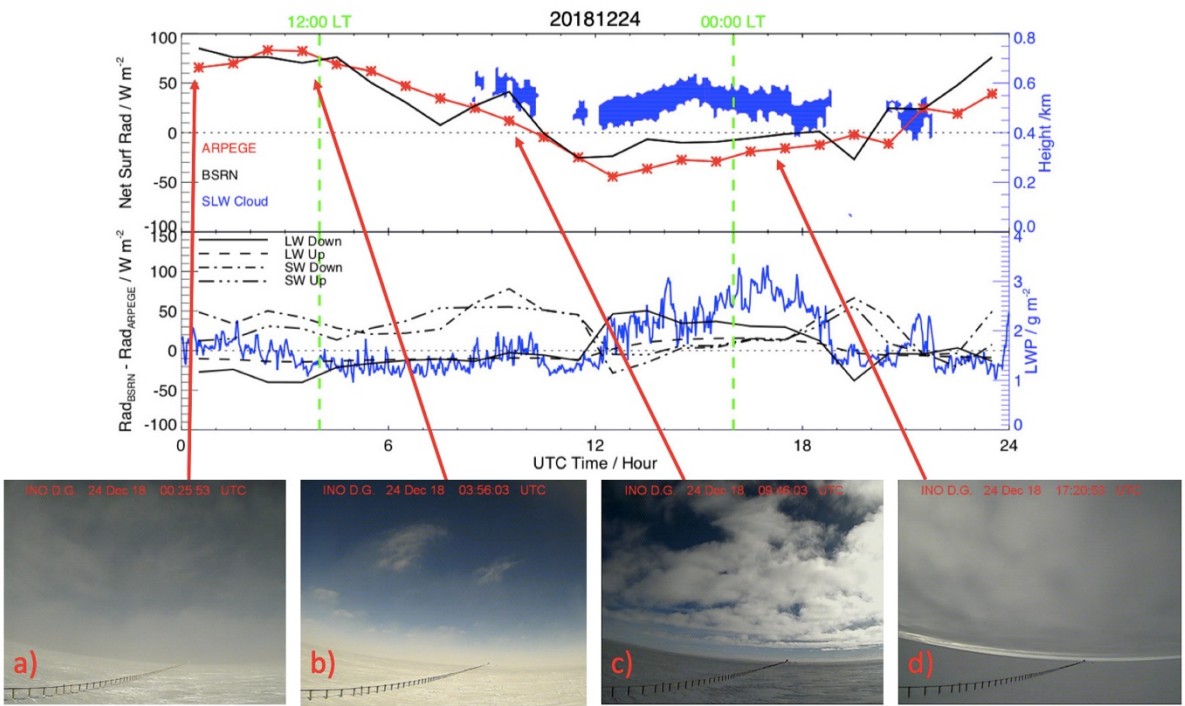

**Figure 15:** (Top) Diurnal variation of the net surface radiation (W m$^{-2}$) observed by BSRN (black solid line) and calculated by ARPEGE-SH (red crossed line) on 24 December 2018 in UTC Time. Superimposed is the SLW cloud height (blue) deduced from the LIDAR. (Middle) Diurnal variation of the difference between surface radiation (W m$^{-2}$) observed by BSRN and calculated by ARPEGE-SH on 24 December 2018 for longwave downward (black solid), longwave upward (black dashed), shortwave downward (black dashed dotted) and shortwave upward (black dashed triple dotted) components. Superimposed is LWP (blue) measured by HAMSTRAD. (Bottom) Four webcam images showing the cloud coverage at: a) 00:25 UTC and b) 03:56 UTC (cirrus clouds, no SLW cloud), c) 09:46 UTC (SLW cloud) and d) 17:20 UTC (SLW cloud). Two vertical green dashed lines indicate 12:00 and 00:00 LT.

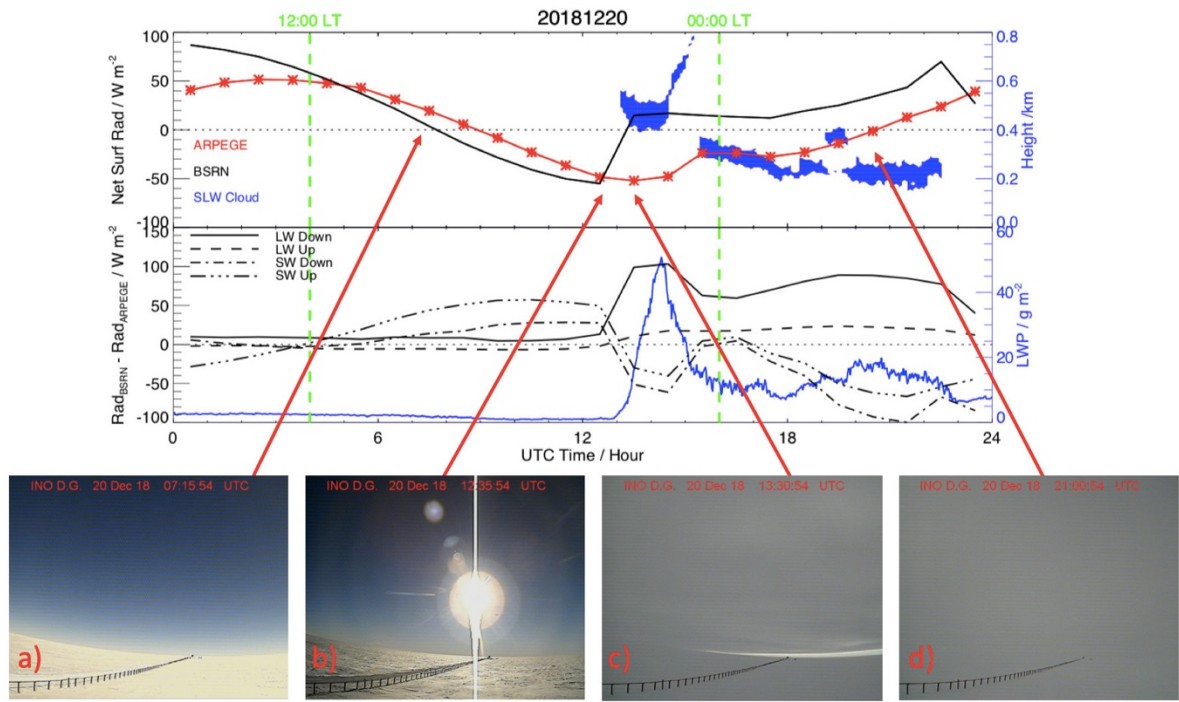


**Figure 16:** Same as Figure 15 but for 20 December 2018 whilst the 4 webcam images were
selected at: a) 07:15 and b) 12:35 UTC (clear sky), c) 13:30 and d) 21:00 UTC (SLW cloud).


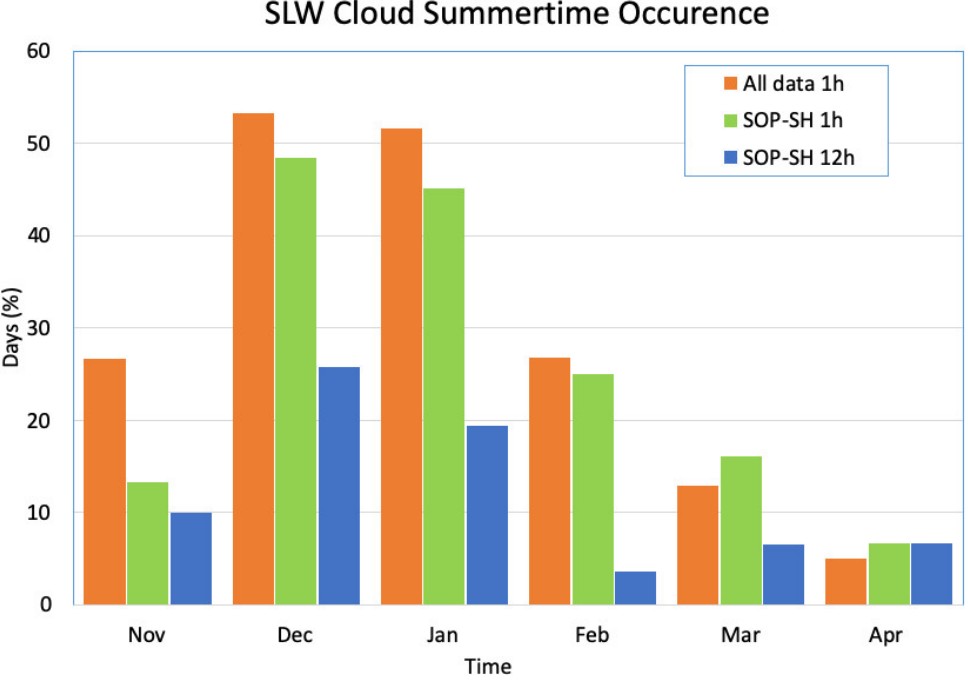


**Figure 17:** Percentage of days per month that SLW clouds were detected within the LIDAR
data for more than 1 hour per day over different summer periods: "All data 1h" (orange) refers
to November (2016-2018), December (2016-2018), January (2018-2019), February (2018-
2019) and March (2018-2019); "SOP-SH 1h" (green) represents the YOPP campaign
(November 2018 to April 2019). "SOP-SH 12h" (blue) represents the percentage of days per
month that SLW clouds were detected during the YOPP campaign within the LIDAR data for
more than 12 hours per day.




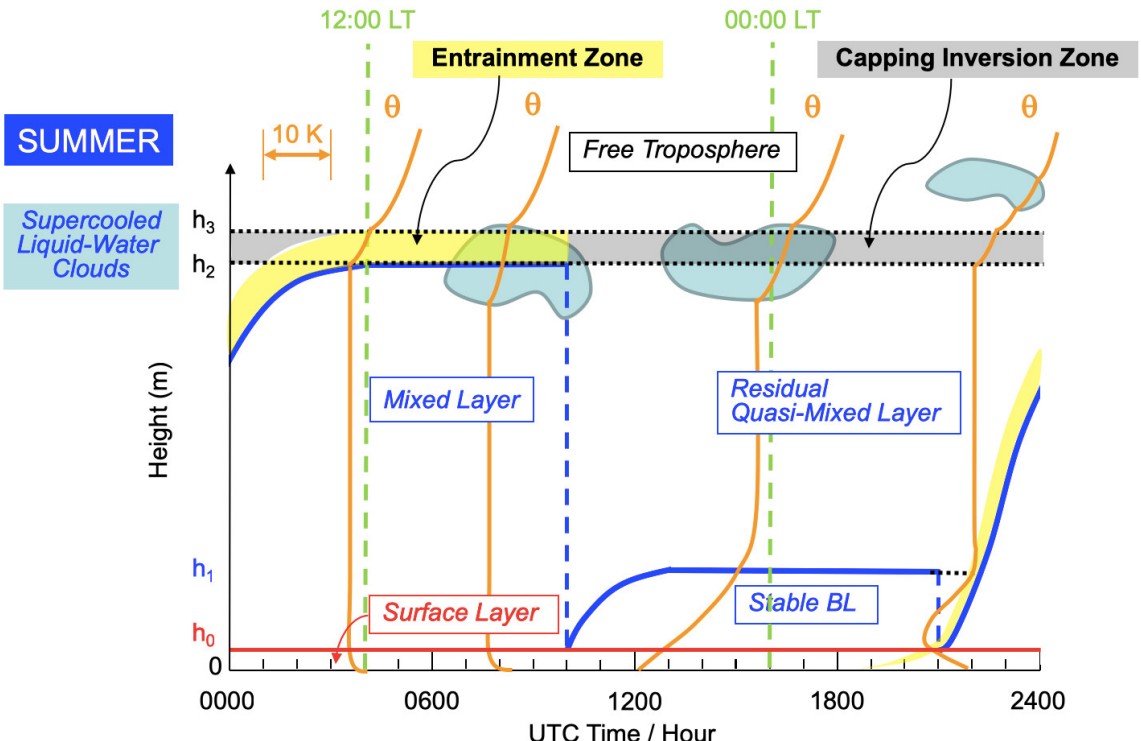


**Figure 18:** Figure modified and updated from Fig. 12 of Ricaud et al. (2012) showing the
diurnal evolution (UTC Time) of the different layers in the Planetary Boundary Layer (PBL)
with h0 the top of the surface layer, h3 the daily overall top of the PBL, and h1 the top of the
intermediate stable layer within the PBL. The orange lines symbolize the vertical profiles of
potential temperature $\theta$, and the light blue areas the SLW clouds. The layer between h2 and h3
is named "capping inversion zone". The yellow area represents the "entrainment zone" at the
top of the (cloudy or cloud-free) mixed layer. When the mixed layer is fully developed, the
entrainment zone coincides with the capping inversion zone. Note that LT = UTC + 8 h,
midnight and noon in the local time reference being indicated by the green dashed lines.

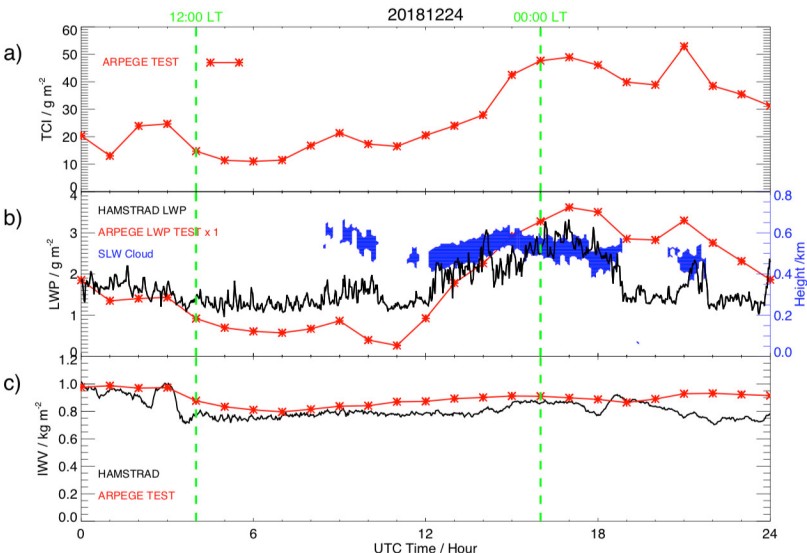


**Figure 19:** Diurnal variation on 24 December 2018 (UTC Time) of: a) the Total Column of Ice
(TCI) (g m⁻²) calculated by ARPEGE-SH in test mode (red crossed line), b) the Liquid Water
Path (LWP) measured by HAMSTRAD (g m⁻², black solid line) and calculated by ARPEGE-
SH in test mode (-no scaling- g m⁻², red crossed line) and c) the Integrated Water Vapour (IWV,
kg m⁻²) measured by HAMSTRAD (black solid line) and calculated by ARPEGE-SH in test
mode (red crossed line). Superimposed to panel b) is the SLW cloud thickness (blue area)
deduced from the LIDAR observations (see Fig. 2c) (blue y-axis on the right of the Figure).
Two vertical green dashed lines indicate 12:00 and 00:00 LT.


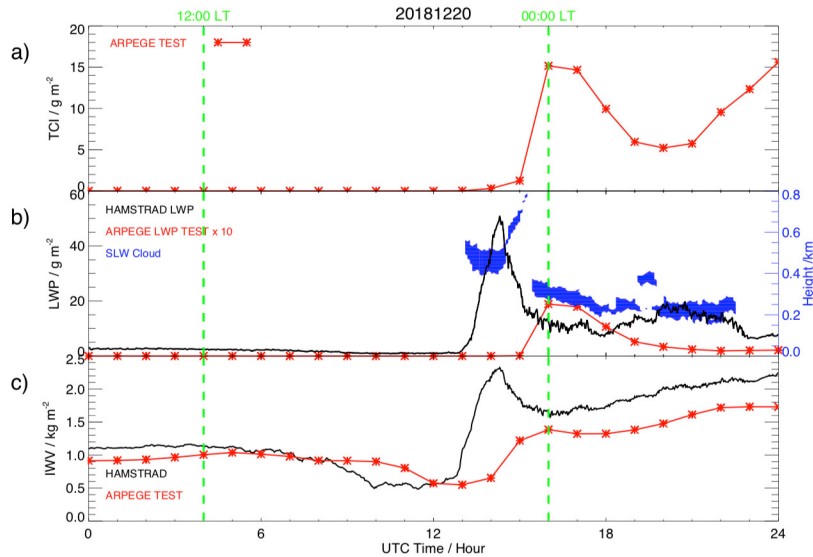


**Figure 20:** Same as Figure 19 but on 20 December 2018 (UTC Time) and LWP from ARPEGE-
SH in test mode has been multiplied by a factor 10.
