# Peer review of "Supercooled Liquid Water Cloud observed, analysed and modelled at the Top of the Planetary Boundary Layer above Dome C, Antarctica"

_Atmospheric Chemistry and Physics, 2019_

## Referee Comment (RC1) · Anonymous Referee #1 · 20 Sep 2019

**Review of "Supercooled Liquid Water Clouds observed and analysed at the top of the Planetary Boundary Layer above Dome C, Antarctica" by Ricaud et al. (acp-2019-607)**
* * *
**Summary:**
* * *
The paper investigates the water budget (cloud, water vapour) in relation to the thermal structure of the boundary layer at Concordia Station, Antarctica. It describes two distinct cases studies from the summer 2018-2019 campaign and highlight the impact of the misrepresentation of supercooled liquid cloud in the ARPEGE model on the surface radiative budget. This study shows that the warmer and wetter episode with cloud leads to radiative biases larger by a few tens of W m-2 than for a more typical configuration of the PBL with colder and drier conditions at "night", when biases of 20-30 W m-2 are already measured. The authors show that this is mainly due to the longwave part of the spectrum, and conclude on the possibly large impact of the misrepresentation of SLW layers on Antarctica's surface energy budget.
* * *
**Relevance of the paper and overall comment:**
* * *
The paper presents very interesting observations of the Antarctic boundary layer, combining cloud, water vapour, and thermodynamic measurements. It clearly demonstrates large biases related to supercooled liquid water (SLW) misrepresentation using a model configuration of ARPEGE (ARPEGE-SH) zoomed in over Dome C. Interestingly, it distinguishs between two PBL regimes, showing that wetter and warmer conditions lead to even larger radiative biases, still related to a misrepresentation of the SLW. To me this dataset allows to address an important question of the link between the modelling of cloud properties (and not just the overall cloud cover) and the surface energy biases measured in Antarctica, which still remain to be 1) understood 2) corrected in NWP models. Moreover, most of the in-situ studies have mostly concentrated on coastal Antarctica so far, and in-situ observations of SLW on the continent are rarely analysed. This study is well in the scope of ACP. I am in favour of its publication in the journal provided improvements are brought to the presentation and discussion of the results. (Minor revision).
* * *
**Main comments:**
* * *
My three main points are:

**1)** Say how the two case studies are representative of the whole summer campaign and please better introduce first the two PBL conditions at once (see my comment on L119 – section 2) and give the synoptic scale context for both cases.

**2)** Provide with a discussion section, which is currently lacking and/or spread over section 3,4,5, in order to provide the reader with a more synthetical views of what is presented in 3 and 4, before concluding in 5.

For instance, you do not discuss the possible bias due to the water vapour (as GHG) vs. the one due to SLW. Given the data-to-model comparison you show, could we say that both are acting as factors biasing the modelled radiations, rather than pointing at SLW only?

Please discuss the fact that the model shows even larger biases in the perturbed (warmer and wetter) case. Is this because the wetter environment allows for a thicker SLW layer to form (hence larger LWP values) and/or also because larger water vapour biases are measured that day? A discussion should compare both case studies.

Can the vertical resolution of the model be responsible for the poor modelling of SLW through failure of simulating enough supersaturation and the right PBL structure? (one would expect higher vertical resolution to allow to better simulate temperature and supersaturation, for instance).

Are there any comments to make regarding instrumental/observations biases from the LIDAR and/or the HAMSTRAD instrument that could somehow affect the conclusions of the study? For instance, the fact that HAMSTRAD is measuring non-null LWP will LIDAR is not seeing any SLW layer in the first case study is interesting. Can the radiometer-derived LWP be biased somehow for instance when large particles of ice precipitate below the SLW layer? (Is this answered in a previous paper e.g. Ricaud et al. 2010b)

Also, have the authors tried to change some settings in ARPEGE in order to allow for more SLW simulation to happen? Is the model not representing SLW because it converts all the vapour into ice or is this rather that it does not even capture the water vapour right? Or both? What can be discussed regarding that matter by comparing both case studies' simulations?

Would you say the radiation biases spotted for the two case studies are representative of the biases for the entire campaign?

I was also wondering whether you were seeing any aerosol with the lidar that could impact the SW radiations, and that would not be considered in the model?

**3)** I recommend to reorganise a little bit the paper by:

- introducing a Method subsection in 2. where you present both types of PBL cases and justify why these two are of particular interest (e.g. representative of most of the campaign?) and give the larger – synoptic – scale context (see comment about L119)

- moving both radiation subsections together in separate (sub)section after the descriptions of both case studies in terms of cloud, temperature, water vapour etc. (see my comment of Line 350 - section 3.4.). In doing so, the main and final aspects of this study (the effect of SLW on radiation budget) would come at once, at the end of the

results section. Both Figures 9 and 16 are very interesting and showing pictures of the cloud cover at the same time is a very good idea.

- adding a discussion section (cf point 2. above)
* * *
**Line by line comments:**
* * *
**Title –** I would rather say Supercooled liquid "layers" (not "clouds") as the examples shown here appear more to be mixed-phase clouds (SLW layer + ice in/below the layer). (see e.g. my comment to L392-393)
* * *
**Abstract**
* * *
L39 –I would start the sentence with "The second case study takes place on..."

L42 and L44 – Since you already said at L31 what you define by a "typical" PBL you do not need these quotation marks here, I think.

L46-48 – I am not convinced by this sentence which is very general compared to the text above and suggests that SLW is absent from all NWPs model over Antarctica, which might to some extent be true, but still this is not shown in the present paper. The verb "indicate" is also not very clear. I would suggest to rewrite this sentence by simply stating that the correct modelling of SLW layers appears crucial to achieve the correct representation of the surface energy budget of the polar atmosphere on the continent.
* * *
**1.Introduction**
* * *
L58 – There are other papers to cite here:

- the Bromwich et al. 2012 cited above

- Listowski, C., Delanoë, J., Kirchgaessner, A., Lachlan-Cope, T., and King, J.: Antarctic clouds, supercooled liquid water and mixed phase, investigated with DARDAR: geographical and seasonal variations, Atmos. Chem. Phys., 19, 6771–6808, https://doi.org/10.5194/acp-19-6771-2019, 2019.

L59 – "(<30%)": This is what Adhikari et al. say in their abstract but please note in winter, when the cloud cover increases over the Plateau, it is more than 30% over at least half of the Plateau (in all of the studies cited above). However, it is indeed less than 30% almost year-round in the area where Dome C sits. You may then want to rephrase a little bit the sentence here.

L63: "…near the coast (Listowski et al. 2019)" Their paper demonstrate this using satellite observations.

L65-66 – The whole Antarctic region, not only the continent I think.

L61 – "Some measurements exist": Yes, and papers that are lacking from the current bibliography investigated the microphysical properties and provided new constraints to modelling. They should be cited here. These are papers dealing with the analysis of airborne measurements:

Grosvenor, D. P., Choularton, T. W., Lachlan-Cope, T., Gallagher, M. W., Crosier, J., Bower, K. N., Ladkin, R. S., and Dorsey, J. R.: In-situ aircraft observations of ice concentrations within clouds over the Antarctic Peninsula and Larsen Ice Shelf, Atmos. Chem. Phys., 12, 11275–11294, https://doi.org/10.5194/acp-12-11275- 2012, 2012.

Lachlan-Cope, T., Listowski, C., and O'Shea, S.: The mi- crophysics of clouds over the Antarctic Peninsula – Part 1: Observations, Atmos. Chem. Phys., 16, 15605–15617, https://doi.org/10.5194/acp-16-15605-2016, 2016.

O'Shea, S. J., Choularton, T. W., Flynn, M., Bower, K. N., Gallagher, M., Crosier, J., Williams, P., Crawford, I., Flem- ing, Z. L., Listowski, C., Kirchgaessner, A., Ladkin, R. S., and Lachlan-Cope, T.: In situ measurements of cloud mi- crophysics and aerosol over coastal Antarctica during the MAC campaign, Atmos. Chem. Phys., 17, 13049–13070, https://doi.org/10.5194/acp-17-13049-2017, 2017.

Also note that Grazioli et al. (2017) observed microphysical properties and shapes of precipitating crystals at DDU (aggregates, rimed particles etc):

Grazioli, J., Madeleine, J.-B., Gallée, H., Forbes, R. M., Gen- thon, C., Krinner, G., and Berne, A.: Katabatic winds diminish precipitation contribution to the Antarctic ice mass balance, P. Natl. Acad. Sci. USA, 114, 1858–10863, https://doi.org/10.1073/pnas.1707633114, 2017b.

L74 – It is also or rather King et al. 2015 that should be cited here, where the authors show the large radiative biases in three high-resolution models and hypothesize the link with the lack of simulated SLW by showing the little liquid amounts formed by these models.

King, J. C., Gadian, A., Kirchgaessner, A., Kuipers Munneke, P., Lachlan-Cope, T. A., Orr, A., Reijmer, C., Broeke, M. R., van Wessem, J. M., and Weeks, M.: Validation of the summertime surface energy budget of Larsen C Ice Shelf (Antarctica) as represented in three high-resolution atmo- spheric models, J. Geophys. Res.-Atmos., 120, 1335–1347, https://doi.org/10.1002/2014JD022604, 2015.

A recent study that should appear in the introduction used the above-mentioned aircraft measurements to specifically show the link between poor/better SLW modelling and poor/better radiation modelling at the surface:

Listowski, C. and Lachlan-Cope, T.: The microphysics of clouds over the Antarctic Peninsula – Part 2: modelling aspects within Polar WRF, Atmos. Chem. Phys., 17, 10195–10221, https://doi.org/10.5194/acp-17-10195-2017, 2017.

These studies above, that deal more with a coastal environment stress even more the importance of the findings of the present paper, which address the continental environment, which has been less investigated so far with respect to links between SLW modelling and radiation biases.

You should also say here that Lawson and Gettelman (2014) conducted a study of SLW observations on the Plateau at South Pole with a MPL. However, if they did look at the radiation changes at the surface by changing some model parameters to simulate more SLW in their model, they did not analyse simultaneous radiation measurement I think (please double-check). You are doing this and this is a big plus of your study compared to theirs in terms of ground-truth radiation budget investigation (and you could emphasise this in your introduction).

L75 – Rather say for instance: "…impacting the radiative budget of the Antarctic and beyond" or something like this, since the Antarctic (including the SO) is the region mainly investigated by Lawson and Gettelman (2014) in their modelling experiments.

L84 – Is there any document or reference describing this project, which could be cited here? Or is the present paper aimed at being the first reference of this project?

L115 – "The method employed and data sets used in our study are…" (see below comment on L119)

L117 – As suggested in my main comment, there should be a discussion section gathering information from 3 and 4 and coming before section 5 (conclusion).
* * *
**2.Datasets**
* * *
L119 – I recommend section 2 explaining not only the dataset used but also the method, i.e. the fact of choosing two scenarios of PBL regimes, and state how representative these two scenarios are for instance (perhaps citing previous work like Ricaud et al. 2012, etc.), with a presentation of both larger synoptic scale contexts. The authors could think of one figure demonstrating the clear difference between the two cases by overplotting temperature e.g. at two different altitides (surface and 500m?) and IWV for both cases. This would better introduce the results for the two case studies. Ideally this would have been event better to compare both atmospheric conditions with the corresponding average+/-std of the whole summer campaign (I am thinking of the representativeness of the two case studies.)

L137 – Does this wet bias takes into account any possible dry bias of the sondes?

L138 – Do you mean studied or validated?

L178 – Please say here which version of the CALIOP product you are using.

L188-189 – what is the vertical resolution of the model configuration, at least in the PBL? 7.5 km stands for the horizontal resolution.

L190 – Since you show cloud fraction in some figures, please recall here how this cloud fraction is defined? What do the values shown in Figure 2a exactly mean?
* * *
**3.Typical diurnal cycle of the PBL**
* * *
L198 – Here you could build on the Method already given in section 2 as recommended in my comment of Line 119, instead of just saying a "typical" PBL cycle, which is not necessary transparent to a reader non familiar with Antarctica.

L203 – "LIDAR cloud backscatter" is redundant with saying that it "indicates that clouds,.." are present. Just say: LIDAR backscatter, and use beta. No need for beta_c

here, I think, as long as you refer to "high values" of beta (clearly defined as >100*beta_mol)).

L211-212 – I recommend rather saying a "SLW layer" because what we would call "cloud" here would rather appear to be the combination of the SLW layer at the top and ice below (and probably in) the SLW layer (hence a mixed-phase cloud). Please change here and everywhere in the text, where relevant.

L214 – See my comment of Figure 2 (end of this review) for the choice of colour, which is not the best here I think.

L218 – I would say: the cloud is mainly confined

L225-227 – The modelled SLW shows very low mmr: 5 10-9 kg/kg = 5 10-6 g/kg ~ 4 10-6 g/m3. Compare this to typical values to Antarctic/Arctic stratus (on coastal areas) of about 0.1 g/m3. (cf. O'Shea et al. 2017, Lachlan-Cope et al. 2016 / Young et al. 2016).

First two were cited above, the third one (as an example) :

Young, G., Jones, H. M., Choularton, T. W., Crosier, J., Bower, K. N., Gallagher, M. W., Davies, R. S., Renfrew, I. A., Elvidge, A. D., Darbyshire, E., Marenco, F., Brown, P. R. A., Ricketts, H. M. A., Connolly, P. J., Lloyd, G., Williams, P. I., Allan, J. D., Taylor, J. W., Liu, D., and Flynn, M. J.: Observed microphysical changes in Arctic mixed-phase clouds when transitioning from sea ice to open ocean, Atmos. Chem. Phys., 16, 13945–13967, https://doi.org/10.5194/acp-16-13945-2016, 2016.

When saying " presence of SLW cloud almost all day long in ARPEGE-SH compared to SLW clouds from 08:00 to 22:00 UTC in the observations" you are not comparing very similar things. The "all day long" SLW layer in the model has very low concentrations, that would – if real at all – be missed by the lidar, especially if it forms above the precipitating ice detected during the first half of the day by the instrument. These very low values should probably be mentioned before comparing things.

Since you speak of the SLW in this paragraph you should also point to the LWP comparison between HAMSTRAD and ARPEGE (Fig. 3b) to give a more quantitative estimate of the difference between both. The sentences of the next paragraph (L228-241) speaking of LWP should be moved to here I think and the next paragraph would only focus on ice and water vapour.

About Fig3b – Interestingly the lidar detects some SLW at 9UTC while HAMSTRAD LWP increases only very slightly but was already non-null before. Why is that? Could it be that at earlier times the lidar is not seeing SLW because of the obscuration by the ice below it (Fig1a)? HAMSTRAD sees non-null LWP values. Are these real? If they are, then you could say that the model is right in continuously simulating SLW (although very small amounts) after all, since HAMSTRAD does continuously detect non-null LWP (as opposed to the LIDAR).

L240-241 – I am not sure why this is recalled here since this was already said before (see my previous comment to line 137).

L242-247 – I would rather say that CALIOP complements (and not validate) the ground observation since it observes from the layer top downwards, and the

ground-based LIDAR from the bottom upwards. Also note that both will not have the same field of view at all so that features seen by the ground based lidar could be missed by CALIOP. The ground lidar will be more prone to detect finer structures and ice below the SLW (since CALIOP signal will get extinguished by the SLW layer). Besides, in the VFM of CALIOP note that SLW is spotted by the space lidar but no ice is detected, while the LIDAR does detect some (Fig. 1a shows that there is ice below the SLW layer). This is most probably due to CALIOP signal getting extinguished because of the presence of SLW. This should probably be commented on in the paper at some point, to help the reader understand the observations.

In Fig4a – Note that the feature detected by CALIOP seems almost to lie on the surface (I am not sure what the green colour is in Fig4a – see my comment about Figure 4 regarding this matter). What is the measured height of this feature compared to the surface? How can we say this is not an artefact? Can SLW missdetections happen very close to the surface within the CALIOP product that is used here?

L251 – How close? How does this distance compare to the typical dimension of the SLW layer in the other direction (the 280km of horizontal extent you mention later). We are not necessarily observing the same layer here, after all.

L261 – same remark about the distance

L264 – I would suggest a title saying "vertical profiles of temperature and water vapour" since you already speak of water vapour in the previous section. The title, for now, suggests that water vapour was not mentioned before.

L291-293 – This explanation about how the PBL is derived should come in section 2 in the subsection dealing with ARPEGE. Actually, the PBL height is already superimposed in all Figures 1a-c and 2a-c without explaining how it was derived.

L295 – Shouldn't you cite King et al. 2006 here?

L300-302 – I am not sure to understand this. What is exactly meant by "elsewhere in the surrounding environment"? Plus, on the figures, the SLW layer after 12UTC seems to remain in a colder (not "warmer") environment (Fig5a and 5b). Then, the model suggests a dryer (not wetter) environment (Fig 5c) and the observation a wetter environment (Fig 5d). I might be missing something here.

L307 – Please define residual mixed layer.

L315-316 – The SLW layer is just below, and not coinciding with, the local max. of dtheta/dz. As it is a bit difficult to see this local maximum on the Figure 6, can you give its height and value in the text?

L317 – I would plot the RS for the potential temp. gradient starting at 100m above the surface, to avoid the unnecessary features/artefacts at the bottom of the red curves.

L323-324 – Since ARPEGE cannot reproduce the fine vertical structure of the theta gradient I would say this as such, instead of saying "broadly consistent", because it seems that this fine structure may in the end be one reason for the wrong simulation

of SLW. To me, just saying "broadly consistent" suggests that this is ok and we don't need to further pay attention to this.

L327 – You speak about colocation. Can you add on Figure 7 a horizontal line giving the height of the SLW layer as obtained from the LIDAR? This would help locate this layer vs. the altitude of the dtheta/dz maximum.

Also I would rather speak of a colocation of the positive dtheta/dz "with the SLW layer", not "with the height of the SLW layer".

However, here, you speak about the colocation of the positive dtheta/dz with the layer while, before, you were rather speaking of the colocation of the maximum of the dtheta/dz with the layer. Do you mean to say both or just one of both? Please remain consistent within this subsection.

L332-333 – Can you recall for the reader the definitions of these zones by e.g. describing a bit more your Figure 8, rather than just referring to previous papers?

L336-348 – In this paragraph you speak about observations over the entire YOPP campaign while it was only about a specific case study so far. This is confusing. Please remove this paragraph. It rather belongs to the discussion section, like the one I am recommending to add in my main comment at the beginning of this review.

L350 - Section 3.4 – I recommend presenting the second case study of SLW layer here. After, you could have a subsection dedicating to surface radiations for both case studies at the same time. It is better not to separate both SLW/PBL case studies so much so that the reader can compare them easily. Surface radiation considerations can very well be moved to a common part, later in the paper and it would be better to show Figures 9 and 16 at the same time, so that – again – both cases can be paralleled.

L353: Figure9 (top)

L365-368. "As the SLW… over Antarctica". This sentence would better go in the discussion section that I am recommending to add. Focus on the case study here. Also, you don't necessarily know whether what CALIOP sees is exactly the cloud you see from the ground. Plus, note that it is 280 km along the satellite track and you don't know about the cloud cover in the perpendicular direction (unless the second orbit you are not showing gives info about cloud cover size along a different direction?).

L370: you only mention the increase in downward LW radiation. In theory you should also detect a decrease in SW because small droplets are very efficient in reflecting sunlight. And, actually, you do see this in your plot. Around 12UTC you see a reduction in down/upward surface SW, because of the SLW layer reflecting sunlight, hence reducing the upward SW (reflection by the icy or snowy surface) as well. Please refer to this as this satisfyingly shows the opposite effects of the SLW layer in both parts of the spectrum. (see my similar comment of L475-476, for Figure 16)

L373: What is meant by "at a level higher"?
* * *
**4.Perturbed diurnal cycle of the PBL**
* * *
L377-380: this info could go in the "method" subsection I was recommending to add, to introduce at once both PBL cases investigated here (and possibly say something about their representativeness).

L392-393 – I am not sure to understand why it is said that the LIDAR does not detect a mixed-phase as well. Your LIDAR mask indicates SLW but this does not mean there is no ice at the same time. Does it? Fig10a suggests ice forms/falls below SLW layer I suppose and again it could be ice precipitating from the SLW layer where little crystals would have also already formed (as it is often the case for low-level mixed-phase clouds). Unless you demonstrate this, I don't think you can say here that your observed cloud is not a mixed-phase one.

L400: the simulated cirrus cloud is above the area where the SLW layer is. This is not just a matter of sensitivity. This is most probably because the SLW layer extinguishes the lidar signals which cannot reach the cirrus cloud. The top right part of Fig10a suggests so (no signal).

L393 – As for the previous case study, please do clearly highlight the very little amounts formed (in g/kg) clearly showing that the model forms virtually no liquid at all...

L401 - SLW layer

L429 - Please avoid the use of "model data". You could say e.g. "the model output" or "the model simulates a moistening…"

L449 - When you say "This is broadly…" it is not clear what "This" is referring to since you then speak about the events "prior the warm episode", while "This" seems to refer to all three profiles. Please rewrite. I am not comfortable with this "broadly consistent" expression (see my previous comment on L323-324). Obs-Model differences in Fig15b and c appear even more larger here than in the previous case study.

L458 - Figure 16 (top)

L472 - the maximum of LWP appears rather to be 45 g m-2 at 13h00 UTC. However, it seems that you smoothed the data here – when comparing with Fig.12b where we see the maximum is 50 g m-2 indeed. Please make Figure/text consistent with each other.

L475-476 - As for the previous case-study you can note the decrease in SW up/down because of the reflection of sunlight by the SLW layer. Compared to previous case, however, SW up/down is continuously decreasing. Why is that? It seems, according to the pictures and the LIDAR detection that the cloud deck is thickening (hence the largest LWP values) and the cloud base lowering, preventing always more radiations to reach the ground, while for the previous case study, the cloud seemed more broken (see your pictures) and the cloud base altitude constant (see the LIDAR detections).

These types of observations should be commented on, here or in the discussion section. Please make the most of the combination of radiation measurements, cloud measurements, and visual observations. Again, rearranging the paper so that Figure 9 and 16 are in a same unique section about radiation would help.

Also, you don't comment on the fact the water vapour bias is clearly appearing in this case study so that you could expect also bias in LW radiations from water vapour since it is a strong GHG. Perhaps this can explain the larger biases observed in the second case study (in addition to the thicker SLW layer observed). What do you think? These are matters to discuss in a discussion part…
* * *
**5.Conclusions**
* * *
L496 - you have not mentioned any CALIPSO overpass for the 20th of December so far, only for the 24th. Both overpasses you were mentioning (although you only showed one) were for the 24th.

L498: underestimated – say by how much.

L505 - BSRN LW or net values?
* * *
**Figures.**
* * *
Figure 1 – one cannot see the red text, especially on the coloured background. Please adapt the color, and put the text a bit higher. For Figure 1c, use the blue colour for the "SLW layer".

Figure 2 – Please use a different color for the text in the plots. Also, this is confusing to have "liquid water" written in red, with that colour being used for the lidar observation as well (while also being part of the colourscale…) Could you perhaps use grey colour to indicate the SLW layer observation? Also: using white colour instead of black-red would help seeing the curve for PBL height more clearly.

Figure 4 – Can you put the names of the categories on the colourbars instead of numbers that are not explained in the Figure caption? If this Figure is a quicklook obtained from another source, this should probably be said.

Figure 5 - Please use a color other than red for text.

Figure 6 - Please use other color for observed SLW as it is the same red as in the colourscale.

Figure 9 – "simulated with" rather than "calculated by"?

Figure 11 – It might be better to use white colour for the PBL height.

Figure 13 - red text is not visible. Also, the red colour chosen to show the presence of the SLW observed by the LIDAR is the same red as the colour scale. Please change this. This was not so problematic in Figure 5 but it is here.

Figure 14 - using red colour for SLW and the same red for the colourscale shoud be avoided.

---

## Referee Comment (RC2) · Anonymous Referee #2 · 7 Oct 2019

**Review of "*Supercooled Liquid Water Clouds observed and analysed at the Top of the Planetary Boundary Layer above Dome C, Antarctica*" by Ricaud et al.**

Ricaud et al. present a very nice study of two cases of supercooled liquid water cloud layers measured using a suite of remote sensing instruments at Dome C, Antarctica in the summer of 2018. Exemplar cases of a "typical" and a perturbed boundary layer are detailed to show what cloud and boundary layer properties may be expected in the region and how these properties can be affected by warm moist oceanic air masses. The authors show that these perturbed boundary layers can greatly change the radiative properties of the clouds which form within them and thus affect the surface energy balance.

Comparisons with the ARPEGE-SH model show that the model fails to capture key observed characteristics of the clouds and boundary layer structure in both scenarios; specifically, the model severely underpredicts cloud supercooled liquid water and exhibits almost systematic biases in temperature and water vapour with respect to the observations. The failure of the model to capture cloud presence and phase distribution the both cases is very important to highlight to the community. The difference between observed and modelled net surface radiation in the perturbed case (up to 50 W/m$^2$) is particularly striking.

The study is well explained and provides clear figures to support the conclusions drawn from the observation-model and inter-case comparisons. It will provide an excellent resource as a reference study for future observation-model comparisons focusing on boundary layers on the Antarctic Plateau. The authors operate with transparency by providing access to all data used within the study, which is fantastic to see. I recommend publication subject to some minor comments and restructuring.

**General comments:**

I appreciate that statistical analyses of PBL properties measured/modelled at Dome C is within the future work remit of this study; however, it would be useful to provide the reader with an estimation of how representative each of these two cases were for e.g. the summer of 2018/19. It would be useful to know how typical is "typical" with perhaps an estimation of occurrence percentage. Also, it would be helpful to have some synoptic overview of the two cases presented, perhaps as a supplementary to the manuscript, with some reference to the mean synoptic state over e.g. the summer of 2018/19.

The authors mention that model 24-h model forecasts and meteorological analyses were provided, where 4D-VAR data assimilation of the latter took place every 6 h. It is stated that most of the model-observation comparison uses the forecasts, while the analyses are used at 0000 UTC and 1200 UTC

(understandably due to radiosonde ingestion improving the comparison and providing a "best guess" at these times). What is unclear to me is the combination of these two datasets, and this should be made clearer in the manuscript. When are the forecasts initialised? Are they initialised every 12-h (0000 UTC or 1200 UTC) or once per day? Are they 24-h in total, or are 24 hours of each forecast (which may be for longer) used for comparison with the observations? Or, are they 24-h in total, initialised twice daily at 0000 UTC or 1200 UTC, providing the latter 12 hours for comparison with the observations (to avoid any spin up issues)?

Indeed, if 12-h subsets of the 24-h forecasts are used, with each beginning at either 0000 UTC or 1200 UTC, then the sharp transitions at 1200 UTC would be somewhat expected from this re-initialisation as the model is effectively brought from maximum divergence back to the "best guess" of the atmospheric state. The authors mention the poor agreement in cloud properties at 1100 UTC and 2300 UTC in Section 3.1, so I am inclined to believe this is how the model is being operated. One would expect improved agreement with the observations at these re-initialisation times (albeit there may still be some discrepancies with the observations which should be emphasised). If I have misunderstood, please accept my apology; however, I feel that the model use and implications of using it in forecasting mode should be discussed in greater detail with a focus on how this re-initialisation (if conducted) may be affecting any of the conclusions drawn with respect to the transitions between wet/dry conditions. Additionally, if this is the case, there is scope for more discussion on model deficiencies: if the model diverges so strongly within the forecast comparison window, it may suggest severe parametrization deficiencies within the model.

Following from this last point, the study would benefit from more discussion on why the model fails to capture the SLW layers, perhaps with specific reference to which cloud microphysical parametrizations are used within the model. Did the authors look into the process parametrizations and identify which may need to be changed to remedy the poor model-observations comparison? E.g. are the deposition freezing / Wegener-Bergeron-Findeisen mechanisms too efficient? To what extent can the SLW deficiency be caused by the poor agreement in temperature / water vapour?

The study is well written; however, it could benefit from a distinct Discussion section for the case and literature comparisons. For example, the 1st paragraph and point (1) of the last paragraph of Section 3.2, 4th and 5th paragraphs of Section 3.3 etc. read like a discussion and should be presented separately from the main study results.

Also, the following additional references could be of benefit to the study:

- O'Shea, S. J., Choularton, T. W., Flynn, M., Bower, K. N., Gallagher, M., Crosier, J., Williams, P., Crawford, I., Fleming, Z. L., Listowski, C., Kirchgaessner, A., Ladkin, R. S., and Lachlan-Cope, T.: In situ measurements of cloud microphysics and aerosol over coastal Antarctica during the MAC campaign, Atmos. Chem. Phys., 17, 13049–13070, doi:10.5194/acp-17-13049-2017, 2017.
- Grosvenor, D. P., Choularton, T. W., Lachlan-Cope, T., Gallagher, M. W., Crosier, J., Bower, K. N., Ladkin, R. S., and Dorsey, J. R.: In-situ aircraft observations of ice concentrations within clouds over the Antarctic Peninsula and Larsen Ice Shelf, Atmos. Chem. Phys., 12, 11275-11294, doi:10.5194/acp-12-11275-2012, 2012.
- Young, G., Lachlan-Cope, T., O'Shea, S. J., Dearden, C., Listowski, C., Bower, K. N., et al. Radiative effects of secondary ice enhancement in coastal Antarctic clouds. Geophysical Research Letters, 46. doi:10.1029/2018GL080551, 2019.
- King, J. C., Gadian, A., Kirchgaessner, A., Kuipers Munneke, P., Lachlan-Cope, T. A., Orr, A., Reijmer, C., Broeke, M. R., van Wessem, J. M., and Weeks, M.: Validation of the summertime surface energy budget of Larsen C Ice Shelf (Antarctica) as represented in three high-resolution atmospheric models, J. Geophys. Res. Atmos., 120, 1335–1347, doi:10.1002/2014JD022604, 2015.

**Specific comments:**

**Page 2, line 29:** Please define ARPEGE-SH as an acronym at first point of use.

**Page 5, line 89:** Please define YOPP SOP as SOP-SH to avoid confusion with SOPs 1-3 in the northern hemisphere. This is described in more detail in Section 2.6, but it would be beneficial to move these specific dates up to this point (or just repeat).

**Page 5, line 95:** Here, it is not clear what the authors mean by the "adjustable time resolution" statement, as it is not clear whether 7 mins was chosen as the time resolution or whether it is the limit of what is achievable by the instrument. However, this becomes clear within the Methods section. Please rephrase for clarity or remove.

**Page 6, line 117:** typo (synthesizes)

**Section 2.1:** can the authors comment of whether you would expect instrument functionality to be affected by the altitude difference between Pic du Midi and Dome C?

**Section 2.2 (and throughout):** The authors often refer to measurements with respect to mean sea level; however, given the high altitude of Dome C, this is misleading to the reader. Please could measurements made at Dome C be rephrased to "above ground level"? This would avoid any confusion.

**Section 2.6:** How many model levels were within the PBL? Can the authors comment on whether the vertical resolution of the model may limit its ability to capture the ~100m thick SLW clouds observed? Additionally, the spatial resolution is quite coarse: Young et al., 2019 (full reference above) found that resolution can affect cloud modelling skill, can the authors comment on whether this may be affecting their comparisons?

**Page 11, lines 222-223:** Is it not only the SLW from the lidar that is shown in Figure 2?

**Page 11, lines 232 – 234:** Small values are presented from the HAMSTRAD, what is the measurement accuracy of this instrument?

**Page 12, line 260:** Second pass of CALIOP/CALIPSO not shown – please consider adding figure in supplementary material to the manuscript.

**Page 12, line 266:** Please include reference edition number for Pruppacher and Klett as figure numbering may change between editions.

**Page 14, lines 313 – 316:** mention that this positive gradient indicates a stable layer, as previously explained. As currently written, the authors are leaving it to the reader to make this conclusion and should be explicitly emphasised.

**Figures:**

- The description of the model BL calculation is included for the first time in the introduction to Figure 5 on page 13. As this BL height is used for the first time in Figure 1, it should be introduced at this first point of use (page 10).
- I would suggest using a different colourmap between observations and modelling results to make it clear to the reader which data are being presented.
- Figure 4: Could benefit from (a) larger tick labels / different labels to explain phase masks rather than providing the number allocation; (b) indicating the altitude of Dome C to illustrate cloud layer altitude relative to ground level.

- For anomaly figures (5 and 13), please consider using a diverging colourmap with white at 0 (e.g. blue-white-red) to ease readability. Changing the colormap may make the sub-figure headings clearer, which would also be useful.

- Figure 7: it may be useful to adapt the scale to make the subtle maxima easier to see.

- Figure 14: it's quite hard to see the measured SLW cloud layer (red) on top of the high values of delta Theta (red), perhaps changing the colour (maybe white?) of the measurements would make this easier to distinguish.

- Side note: Figures 9 and 16 are fantastic, the webcam images do a great job at illustrating the different cloud conditions between the two cases. The radiation values alone don't truly convey how different cloud distribution can be, so these images are invaluable to emphasise this.

---

## Author Comment (AC1) · 19 Dec 2019

**Version 6, 19 December 2019**

**Manuscript Title:** Supercooled Liquid Water Clouds observed and analysed at the Top of the Planetary Boundary Layer above Dome C, Antarctica **by Ricaud et al.**

**RESPONSES TO THE EDITOR**

 $\rightarrow$  Both reviewers requested structural changes to the paper, as well as provided line edits. The line edits were made before large passages were moved around. In response to suggestions from the reviewers, a section focused on the impact of the SLW clouds has been created as well as a Discussion section. As suggested by the reviewers, we have created a supplementary file where additional information has been inserted. Specific changes have been made in response to the reviewers' comments and are described below. The reviewers' comments are recalled in blue and changes in the revised version are highlighted in yellow. We have acknowledged the two anonymous reviewers. A sentence has been inserted in the Acknowledgements.

We would like to thank the two anonymous reviewers for their beneficial comments.

Note that Figures and Table are labelled as follows:

Figs. 1-18: Figures shown in the revised manuscript Figs. Supp1-Supp14: Figures shown in the Supplementary Materials Figs. R1-R4: Figures only shown in the Replies to the Reviewers Table R1: Table only shown in the Replies to the Reviewers

**Anonymous Referee #1**

**Review of "Supercooled Liquid Water Clouds observed and analysed at the top of the Planetary Boundary Layer above Dome C, Antarctica" by Ricaud et al. (acp-2019-607)**

 $\rightarrow$  Both reviewers requested structural changes to the paper, as well as provided line edits. The line edits were made before large passages were moved around. In response to suggestions from the reviewers, a section focused on the impact of the SLW clouds has been created as well as a Discussion section. As suggested by the reviewers, we have created a supplementary file where additional information has been inserted. The reviewers' comments are recalled in blue and changes in the revised version are highlighted in yellow. In the following responses, note that Figures and Table are labelled as follows:

Figs. 1-18: Figures shown in the revised manuscript Figs. Supp1-Supp14: Figures shown in the Supplementary Materials Figs. R1-R4: Figures only shown in the Replies to the Reviewers Table R1: Table only shown in the Replies to the Reviewers

**Summary:**

The paper investigates the water budget (cloud, water vapour) in relation to the thermal structure of the boundary layer at Concordia Station, Antarctica. It describes two distinct cases studies from the summer 2018-2019 campaign and highlight the impact of the misrepresentation of supercooled liquid cloud in the ARPEGE model on the surface radiative budget. This study shows that the warmer and wetter episode with cloud leads to radiative biases larger by a few tens of W m-2 than for a more typical configuration of the PBL with colder and drier conditions at "night", when biases of 20- 30 W m-2 are already measured. The authors show that this is mainly due to the longwave part of the spectrum, and conclude on the possibly large impact of the misrepresentation of SLW layers on Antarctica's surface energy budget.

**Relevance of the paper and overall comment:**

The paper presents very interesting observations of the Antarctic boundary layer, combining cloud, water vapour, and thermodynamic measurements. It clearly demonstrates large biases related to supercooled liquid water (SLW) misrepresentation using a model configuration of ARPEGE (ARPEGE-SH) zoomed in over Dome C. Interestingly, it distinguishes between two PBL regimes, showing that wetter and warmer conditions lead to even larger radiative biases, still related to a misrepresentation of the SLW. To me this dataset allows to address an important question of the link between the modelling of cloud properties (and not just the overall cloud cover) and the surface energy biases measured in Antarctica, which still remain to be 1) understood 2) corrected in NWP models. Moreover, most of the in-situ studies have mostly concentrated on coastal Antarctica so far, and in-situ observations of SLW on the continent are rarely analysed. This study is well in the scope of ACP. I am in favour of its publication in the journal provided improvements are brought to the presentation and discussion of the results. (Minor revision).

 $\rightarrow$  Thank you for your positive comments.

**Main comments:**

**My three main points are:**

1) Say how the two case studies are representative of the whole summer campaign and please better introduce first the two PBL conditions at once (see my comment on L119 – section 2) and give the synoptic scale context for both cases.

→ Based on the NCEP data sets, the temperature fields at 600 hPa above Antarctica have been investigated both during the two case studies and climatologically during the YOPP campaign (December 2018-January 2019) and over 10 years in summer (December-January) from 2009 to 2019 (Figure 1). Climatologically the Dome C station temperature at 600 hPa is less than 245 K. This is consistent with the temperature analysed on 24 December 2018 during the case study labelled as "typical". On 20 December 2018 (case study labelled as "perturbed"), warm air parcels (temperature greater than 260 K) are issued from the coast opposite in longitude (30°W) of the Dome C station creating an elongated tongue of warm air (temperature greater than 250 K) with maxima of 255 K on 21 December 2018 at 00:00 UTC above Dome C.

---

## Referee Report (RR1)

**Review of the revised version of "Supercooled Liquid Water Clouds observed and analysed at the top of the Planetary Boundary Layer above Dome C, Antarctica" by Ricaud et al. (acp-2019-607)**
* * *
**Main comment:**
* * *
I am very satisfied with the answers and improvements made by the authors to the manuscript. The two case studies are clearly put into context and the authors assess how representative they are. The discussion is definitely a plus and improves the paper. To me the essential aspects are addressed and the paper can be published almost as is, but I list a few points below that should be considered in my opinion.

- Why not showing figures of the modeling experiments in the main text? This is important and interesting to show that the typical case can be simulated by changing microphysical parameters (the partition scheme), while this is not enough for the perturbed case. (please see also my comment of Lines 742-765).
- Please mitigate a bit the SLW cloud vs Mixed-Phase Cloud discussion (please see my comment of the section 7.1)
* * *
**Line by line comments:**
* * *
**Title –** Considering the substantial effort made by authors to include some modeling experiments by changing the ice/liquid partition scheme, I would tend to suggest this title:

"Supercooled Liquid Water Clouds observed, analysed and modeled at the top of the Planetary Boundary Layer above Dome C, Antarctica"

This is also a paper about modeling, particularly in its revised version.
* * *
**Abstract**
* * *
L32 "…exhibited SLW clouds": please add "for at least one hour"
* * *
**1.Introduction**
* * *
L76: "…sea ice production of ice-condensation nuclei". This is more appropriate to say INPs for Ice Nucleating Particles. However, here, this is also sea ice as a source of CCN and not only INP, which is discussed in the papers. Sea ice could bring CCN in

the form of sea salt. Moreover, you are citing Legrand et al. 2016's paper, which I think is about sea salt measurements coming from sea ice. Sea salt is a good CCN, not a good INP, at these temperatures.
* * *
**4.Typical diurnal case of the PBL**
* * *
L 322: you defined LT but are using LST here. Please define.

L353-361: you could also say that ARPEGE is missing the precipitating ice as well (comparing Figure 2 and 3) between 0 and 12UTC.

L452-465: Two almost identical paragraphs here. I guess you want to keep the second one only.
* * *
**7. DIscussion**
* * ** * *
**7.1 SLW Clouds vs Mixed-Phase Clouds**
* * *
I see your point. However, when we look at Figures 2a and 9a there is a clear sign – to me – of ice (the streaks) coming from the height where the SLW layer is. It is fine to say that the SLW layer is virtually only liquid but it is difficult to think that no process is taking place that depletes liquid and turns in into ice, hence giving the ice seen below the SLW layer. I think it is difficult to rule out the fact that the SLW layer is not a part from a – microphysically speaking – mixed-phase process. Where would the precipitation come from then? Small (undetected) crystals can be falling out of the SLW layer then grow while they fall so that the lidar detects them, eventually.

Put it more simply I can see why saying that the "ice component, even if present, is irrelevant from a radiative point of view" (Line 658-659). I think it is not from a microphysical point of view. The authors recognize themselves that there might by small crystals not seen by the lidar ("Some S signal is nevertheless present…" L644-645). How do you know the SLW layer is not slowly disappearing also because it is slowly converted into ice, which precipitates (see e.g the end of the day Figure 9a)? In which case the ice microphysics would also be important since it guides the termination of the SLW layer, hence impacting (also!) the radiative budget (indirectly).

Unless the authors have something against this argument, I would like to see something about this in the discussion to mitigate the "SLW cloud". To me you are investigating the SLW layer of an overall mixed-phase process (or say mixed-phase cloud).

More generally it is always difficult to say what is and what is not a mixed-phase cloud and definitions differ especially when observing with different instruments (space lidar vs. ground-based one, for instance).

I think here, it is not entirely fair to the complexity of mixed-phase microphysical processes to treat these SLW layers as "pure" SLW clouds. It just rules out the role ice might be playing in driving their lifetime (hence indirectly impacting the radiative budget).
* * *
**7.3 SLW Clouds in ARPEGE-SH**
* * *
L722:
-Then it would be interesting – if easily doable ? – to know how frequent SLW layer with a lifetime >12h (and not >1h) are, so that the first case's representativity (in terms of SLW layer lifetime) could also be assessed. What does Figure 17 with a 12h- and not 1h-criterium give? This would back up more the "may have a strong impact on the calculation of the radiation budget" (Line 723).

-There is not any section 3.1, only a section 3, now.

L733: "… since the cloud water is not a model control variable in the 4DVar scheme, it cannot be analysed"

I am not sure what is meant here by just "analysed": do you mean the analysis step of the data assimilation process? or just the fact of analysing an output? I suppose it is the former and then I would advise to say: "updated by the analysis step of the 4DVar data assimilation process".

L742-765: I don't see why you would not show the figures with the modeling experiments here (Fig. Supp11 and Supp12), in the main text. You can very well leave the Figure showing the partition scheme in the Supp. Material, however. This is very interesting to see that you manage to find a partition scheme, which improves a lot the modeling for the first case at least, but also is not enough to solve the problem (2nd case). In the second case (Figure Supp 12), it clearly seems that the sudden increase of water vapour (advection?) is not reproduced in the model and should be the reason of the only slight improvement in the SLW modeling brought by the new partition scheme.
* * *
**8.Conclusions**
* * *
L779 : recall here "for at least one hour".

---

## Author Response (AR2)

**Manuscript Title:** *Supercooled Liquid Water Clouds observed and analysed at the Top of the Planetary Boundary Layer above Dome C, Antarctica* **by Ricaud et al.**

**RESPONSES TO THE EDITOR**

→ One reviewer requested minor changes to the paper. Specific changes have been made in response to the reviewer's comments and are described below. The reviewer's comments are recalled in blue and changes in the revised version are highlighted in yellow.

As requested by the reviewer, we have modified the title into:

```
Supercooled  Liquid  Water  Clouds  observed, analysed and
modelled at the Top of the Planetary Boundary Layer above
Dome C, Antarctica
```

**Anonymous Referee**

**Review of the revised version of "Supercooled Liquid Water Clouds observed and analysed at the top of the Planetary Boundary Layer above Dome C, Antarctica" by Ricaud et al. (acp-2019-607)**

----------------------- **Main comment:** -----------------------

I am very satisfied with the answers and improvements made by the authors to the manuscript. The two case studies are clearly put into context and the authors assess how representative they are. The discussion is definitely a plus and improves the paper. To me the essential aspects are addressed and the paper can be published almost as is, but I list a few points below that should be considered in my opinion.

→ Thank you

• Why not showing figures of the modeling experiments in the main text? This is important and interesting to show that the typical case can be simulated by changing microphysical parameters (the partition scheme), while this is not enough for the perturbed case. (please see also my comment of Lines 742-765).

→ Modified, see below.

• Please mitigate a bit the SLW cloud vs Mixed-Phase Cloud discussion (please see my comment of the section 7.1)

→ Modified, see below.

----------------------------------- **Line by line comments:** -----------------------------------

Title – Considering the substantial effort made by authors to include some modeling experiments by changing the ice/liquid partition scheme, I would tend to suggest this title:

"Supercooled Liquid Water Clouds observed, analysed and modeled at the top of the Planetary Boundary Layer above Dome C, Antarctica"

This is also a paper about modeling, particularly in its revised version.

→ The title has been modified accordingly.

> Supercooled Liquid Water Clouds observed, analysed and modelled at the Top of the Planetary Boundary Layer above Dome C, Antarctica

---------------------- **Abstract** ----------------------

L32 "…exhibited SLW clouds": please add "for at least one hour"

→ Done

------------------------ **1. Introduction** ------------------------

L76: "…sea ice production of ice-condensation nuclei". This is more appropriate to say INPs for Ice Nucleating Particles. However, here, this is also sea ice as a source of CCN and not only INP, which is discussed in the papers. Sea ice could bring CCN in the form of sea salt. Moreover, you are citing Legrand et al. 2016's paper, which I think is about sea salt measurements coming from sea ice. Sea salt is a good CCN, not a good INP, at these temperatures.

→ We clarified this point and modified the incriminated sentence into:

> These studies also highlighted sea-ice production of Cloud-Condensation Nuclei and Ice Nucleating Particles, which is important in winter both coastally and at Dome C (see e.g. Legrand et al., 2016).

--------------------------------------------- **4. Typical diurnal case of the PBL** ------------------------------------------------

L 322: you defined LT but are using LST here. Please define.

→ We changed LST into LT in all occurrences.

L353-361: you could also say that ARPEGE is missing the precipitating ice as well (comparing Figure 2 and 3) between 0 and 12UTC.

→ This is an interesting comment. Based simply on comparing Figs. 2 and 3, we cannot state any conclusions on the precipitating ice from ARPEGE-SH since only clouds (fraction, ice and liquid) are represented in Figure 3. The diurnal variation along the vertical of the Total Snow Flux (mm day$^{-1}$) calculated by ARPEGE-SH on 24 December 2018 and on 20 December 2018 is shown on Figures Supp2 and Supp3, respectively. Note that ARPEGE-SH computes only solid and liquid phases for precipitating water and cloud. There is no distinction between snow and graupel. On 24 December 2018 (Fig. Supp2), ARPEGE-SH forecasts some solid precipitation between 00:00 and 10:00 UTC from 500 m agl to the surface consistently with the LIDAR observations (Figs. 2a and b). On 20 December 2018 (Fig. Supp3), ARPEGE-SH calculates some traces of solid precipitation close to the surface around 16:00 UTC consistently with the LIDAR observations (Figs. 9a and b). In conclusion, ARPEGE-SH was able to forecast solid precipitation during the 2 case studies. We have inserted the 2 new Figures (Supp2 and Supp3) in the Supplementary Material document and have updated the Figure numbering. We have inserted a new paragraph in the main document.

> The diurnal variation along the vertical of the Total Snow Flux (mm day-1) calculated by ARPEGE-SH on 24 December 2018 and on 20 December 2018 is shown on Figures Supp2 and Supp3, respectively. On 24 December 2018 (Fig. Supp2), ARPEGE-SH forecasts some solid precipitation between 00:00 and 10:00 UTC from ~500 m agl to the surface consistently with the LIDAR observations (Figs. 2a and b). On 20 December 2018 (Fig. Supp3), ARPEGE-SH calculates trace amounts of solid precipitation close to the surface around 16:00 UTC consistently with the LIDAR observations (Figs. 9a and b).

ARPEGE-SH was thus able to forecast solid precipitation during the 2 case studies.

[Figure]

**Figure Supp2:** Time-height cross section on 24 December 2018 (UTC Time) of the Total Snow Flux (mm day$^{-1}$) calculated by the ARPEGE-SH model. Superimposed is the top of the Planetary Boundary Layer calculated by the ARPEGE-SH model (black-white thick line) and the SLW cloud (grey area) deduced from the LIDAR observations (see Fig. 1c). Two vertical green dashed lines indicate 12:00 and 00:00 LT.

[Figure]

**Figure Supp3:** Same as Figure Supp2 but on 20 December 2018.

L452-465: Two almost identical paragraphs here. I guess you want to keep the second one only.

→ Yes, we have removed the first incriminated paragraph.

-------------------------------------------- **7. Discussion** --------------------------------------------

-------------------------------------------- **7.1 SLW Clouds vs Mixed-Phase Clouds** --------------------------------------------------------

I see your point. However, when we look at Figures 2a and 9a there is a clear sign – to me – of ice (the streaks) coming from the height where the SLW layer is. It is fine to say that the SLW layer is virtually only liquid but it is difficult to think that no process is taking place that depletes liquid and turns in into ice, hence giving the ice seen below the SLW layer. I think it is difficult to rule out the fact that the SLW layer is not a part from a – microphysically speaking – mixedphase process. Where would the precipitation come from then? Small (undetected) crystals can be falling out of the SLW layer then grow while they fall so that the lidar detects them, eventually.

Put it more simply I can see why saying that the "ice component, even if present, is irrelevant from a radiative point of view" (Line 658-659). I think it is not from a microphysical point of view. The authors recognize themselves that there might by small crystals not seen by the lidar ("Some S signal is nevertheless present…" L644-645). How do you know the SLW layer is not slowly disappearing also because it is slowly converted into ice, which precipitates (see e.g the end of the day Figure 9a)? In which case the ice microphysics would also be important since it guides the termination of the SLW layer, hence impacting (also!) the radiative budget (indirectly).

Unless the authors have something against this argument, I would like to see something about this in the discussion to mitigate the "SLW cloud". To me you are investigating the SLW layer of an overall mixed-phase process (or say mixed-phase cloud).

More generally it is always difficult to say what is and what is not a mixed-phase cloud and definitions differ especially when observing with different instruments (space lidar vs. ground-based one, for instance).

I think here, it is not entirely fair to the complexity of mixed-phase microphysical processes to treat these SLW layers as "pure" SLW clouds. It just rules out the role ice might be playing in driving their lifetime (hence indirectly impacting the radiative budget).

→ We understand the reviewer's point of view. Other arguments for the presence of a mixed-phase cloud can be proposed when considering precipitating ice detected by the LIDAR and calculated by the model (Figures 2b and Supp2, respectively) on 24 December 2018 between 00:00 and 10:00 UTC (vertical stripes). We have suppressed the sentence that can be considered too much favourable to a pure SLW cloud layer (L. 658: "The layer is thus a truly SLW layer, being that its ice component, even if present, is irrelevant from a radiative point of view."). We have enlarged the discussion to state that the presence of mixed-phase clouds cannot be ruled out.

> On the other hand, when we consider the aerosol depolarization ratio measured by the LIDAR (Figure 2b) and the total snow flux calculated by ARPEGE-SH (Figure Supp2) on 24 December 2018, it is obvious that precipitating ice is present from 00:00 to 10:00 UTC in a layer from ~500 m to the surface (vertical stripes). Therefore, physical processes are supposed to take place within the cloud to deplete liquid and turn it into ice, giving the ice observed and calculated below the SLW layer. In which case, the ice microphysics would also be important since it guides the termination of the SLW layer, hence indirectly impacting the radiative budget. As a consequence, we cannot completely rule out we are investigating a SLW layer of an overall mixed-phase cloud.

--------------------------------------------- 7.3 SLW Clouds in ARPEGE-SH -----------------------------------------------------------------

L722:

-Then it would be interesting – if easily doable ? – to know how frequent SLW layer with a lifetime >12h (and not >1h) are, so that the first case's representativity (in terms of SLW layer lifetime) could also be assessed. What does Figure 17 with a 12h- and not 1h-criterium give? This would back up more the "may have a strong impact on the calculation of the radiation budget" (Line 723).

→ As requested by the reviewer, we have updated the Figure 17 by further considering the percentage of days per month that SLW clouds were detected within the LIDAR data for more than 12 hours per day. A day is flagged with a SLW cloud occurrence when a SLW cloud has been detected in the LIDAR observations for a period longer than 1 hour (orange/green) or 12 hours (blue). As expected, the occurrence of SLW clouds with a lifetime greater than 12 hours (blue) during SOP-SH is less than the occurrence of SLW clouds with a lifetime greater than 1 hour (green). But, whatever the criterium used (1 hour or 12 hours), the maxima of occurrences are shown to be in December and January during SOP-SH. SLW clouds with a lifetime greater than 12 hours occurred about a quarter of the days (20-25%) against about half of the days for SLW clouds with a lifetime greater than 1 hour (40-45%). The occurrence of SLW clouds above the Dome C station in summer is thus a significant phenomenon that may have a strong impact on the calculation of the radiation budget as stated farther in the manuscript. We have inserted a new paragraph.

> In Figure 17, we show the percentage of days per month that SLW clouds were detected within the LIDAR data for more than 12 hours per day (blue) during SOP-SH. As expected, SLW clouds occur less often when they last 12 hours (blue) than when they last 1 hour (green). But, whatever the criterium used (1 hour or 12 hours), the maxima of SLW cloud presence occur in December and January during SOP-SH. 12-h SLW clouds occurred about a quarter of the days (20-25%) compared to roughly half of the days for 1-h SLW clouds (40-45%). This reinforces the argument of the critical importance of well representing SLW clouds in models in order to better estimate radiation budget over Antarctica.

[Figure]

> **Figure 17:** Percentage of days per month that SLW clouds were detected within the LIDAR data for more than 1 hour per day over different summer periods: "All data 1h" (orange) refer to November (2016-2018), December (2016-2018), January (2018-2019), February (2018-2019) and March (2018-2019); "SOP-SH 1h" (green) represents the YOPP campaign (November 2018 to April 2019). "SOP-SH 12h" (blue) represents the percentage of days per month that SLW clouds were detected during the YOPP campaign within the LIDAR data for more than 12 hours per day.

-There is not any section 3.1, only a section 3, now.

→ A careful examination of the paragraph shows that it should be referred not to subsection 3.1 but to subsection "4.1 Clouds" within the section "4. Typical diurnal cycle of the PBL". We have modified the subsection number accordingly.

But the incriminated sentence was mentioning a cloud extension not consistent with the revised version. It was previously written: "As the SLW cloud horizontal extent in the first case study is about 280 km and persists over more than 12 hours (section 3.1), (…)". We modified the sentence into:

> As the SLW cloud horizontal extent in the first case study is between ~450 and ~700 km and persists over more than 12 hours (section 4.1), (…)

For a sake of consistency, we have also updated the conclusion. Since there are no spaceborne LIDAR observations available on 24 December at the time SLW clouds are observed at Dome C, we changed the original sentence "Spaceborne lidar observations revealed horizontal extensions of these clouds as large as 280 and 550 km for the 24 and 20 December cases, respectively." into:

> Spaceborne LIDAR observations revealed horizontal extensions of these clouds as large as 700 km for the 24 December case study.

L733: "… since the cloud water is not a model control variable in the 4DVar scheme, it cannot be analysed"

I am not sure what is meant here by just "analysed": do you mean the analysis step of the data assimilation process? or just the fact of analysing an output? I suppose it is the former and then I would advise to say: "updated by the analysis step of the 4DVar data assimilation process".

→ As suggested by the reviewer, the incriminated sentence has been updated as follow:

> The underestimation of the SLW in ARPEGE-SH can be explained by the fact that: 1) the underestimation of liquid water is mainly a physical problem in the model related to the ice/liquid partition function vs temperature (see below) and 2), since the cloud water is not a model control variable in the 4DVar scheme, it cannot be updated by the analysis step of the 4DVar data assimilation process.

L742-765: I don't see why you would not show the figures with the modeling experiments here (Fig. Supp11 and Supp12), in the main text. You can very well leave the Figure showing the partition scheme in the Supp. Material, however. This is very interesting to see that you manage to find a partition scheme, which improves a lot the modeling for the first case at least, but also is not enough to solve the problem (2nd case). In the second case (Figure Supp 12), it clearly seems that the sudden increase of water vapour (advection?) is not reproduced in the model and should be the reason of the only slight improvement in the SLW modeling brought by the new partition scheme.

→ As suggested by the reviewer, we have inserted in the main manuscript the 2 Figures (Supp11 and 12) initially in the supplementary material. We have modified the text of the main manuscript and of the supplementary material in order to take into account the changes in the numbering of the Figures.

```
For 24 December 2018, and consistently with Fig. 3, we have
drawn on Fig. Supp9 the diurnal evolutions of different
variables calculated by ARPEGE-SH-TEST: a) the Cloud
Fraction, b) the Ice Water mixing ratio and c) the Liquid
Water mixing ratio. Similarly, and consistently with Fig. 4,
Figure 19 presents:  a) the ARPEGE-SH-TEST TCI, b) the LWP
measured by HAMSTRAD and calculated by ARPEGE-SH-TEST and c)
the IWV measured by HAMSTRAD and calculated by ARPEGE-SH-
TEST. Eventually, and consistently with Fig. 9, Figure Supp13
presents the net surface radiation observed by BSRN and
calculated by ARPEGE-SH-TEST, and the difference between
surface radiation of longwave downward, longwave upward,
shortwave downward and shortwave upward components observed
by BSRN and calculated by ARPEGE-SH-TEST. In the same manner,
for the case of 20 December 2018, Figs. Supp12, 20 and Supp14
echo Figs. 11, 12 and 16, respectively.
```

-------------------------------------------- **8. Conclusions** --------------------------------------------

L779: recall here "for at least one hour".

→ Done

---

## Author Response (AR3)

**Version 03.R2, 6 March 2020**

**Manuscript Title:** *Supercooled Liquid Water Clouds observed, analysed and modelled at the Top of the Planetary Boundary Layer above Dome C, Antarctica* **by Ricaud et al.**

**RESPONSES TO THE EDITOR**

Dear authors,
Thanks for the further revisions. I am pleased to accept your manuscript for publication. It will go through language copy-editing. Also as part of improving the language, I suggest a small edit on page 31:

Consider changing "SLW clouds occur less often when they last 12 hours (blue) than when they last 1 hour (green)" to "SLW clouds with a minimum duration of 12 hours (blue) occur less often than SLW clouds with a minimum duration of 1 hour (green)".

Best regards,
Corinna Hoose

→ Thank you very much. As you suggested, we have modified the incriminated sentence in the manuscript V03 file.

> As expected, SLW clouds with a minimum duration of 12 hours (blue) occur less often than SLW clouds with a minimum duration of 1 hour (green).